# A unified mechanism for innate and learned visual landmark guidance in the insect central complex

**Roman Goulard** [1]*, **Cornelia Buehlmann** [2], **Jeremy E. Niven** [2], **Paul Graham** [2], **Barbara Webb** [1]

**1** Institute for Perception, Action, and Behaviour, School of Informatics, University of Edinburgh, Edinburgh, Scotland, United Kingdom, **2** School of Life Sciences, University of Sussex, John Maynard Smith Building, Falmer, Brighton, United Kingdom

* romangoulard@gmail.com

## Abstract

Insects can navigate efficiently in both novel and familiar environments, and this requires flexiblity in how they are guided by sensory cues. A prominent landmark, for example, can elicit strong innate behaviours (attraction or menotaxis) but can also be used, after learning, as a specific directional cue as part of a navigation memory. However, the mechanisms that allow both pathways to co-exist, interact or override each other are largely unknown. Here we propose a model for the behavioural integration of innate and learned guidance based on the neuroanatomy of the central complex (CX), adapted to control landmark guided behaviours. We consider a reward signal provided either by an innate attraction to landmarks or a long-term visual memory in the mushroom bodies (MB) that modulates the formation of a local vector memory in the CX. Using an operant strategy for a simulated agent exploring a simple world containing a single visual cue, we show how the generated short-term memory can support both innate and learned steering behaviour. In addition, we show how this architecture is consistent with the observed effects of unilateral MB lesions in ants that cause a reversion to innate behaviour. We suggest the formation of a directional memory in the CX can be interpreted as transforming rewarding (positive or negative) sensory signals into a mapping of the environment that describes the geometrical attractiveness (or repulsion). We discuss how this scheme might represent an ideal way to combine multisensory information gathered during the exploration of an environment and support optimal cue integration.

## Author summary

In this paper, we modeled the neural pathway allowing insects to perform landmark guided behaviours using their internal compass. First, by copying available details of the neural connectivity between internal compass neurons and steering neurons in the fruit-fly brain, we show this circuit can produce directed behaviour towards a visual landmark. We then propose a mechanism by which this connectivity could be adapted through

**Data Availability Statement:** Data and code available from the following repository: https:// github.com/RomanGoulard/ModelCX_code-data.

**Funding:** We acknowledge support from the Biotechnology and Biological Sciences Research Council (BB/R005036/1). RG and CB received salary from BB/R005036/1. The funders had no role in study design, data collection and analysis, decision to publish, or preparation of the manuscript.

**Competing interests:** The authors have declared that no competing interests exist.

experience to support flexible landmark guidance behaviours such as attraction or menotaxis, that is, movement in arbitrary directions relative to the landmark. This mechanism allows a simple goodness/badness signal, from innate or long-term memory pathways, to be converted into an oriented steering signal relative to the visual surroundings. Furthermore, by simulating lesion experiments in the mushroom bodies of wood ants we highlight the consistency of the model with biological observations.

## 1 Introduction

An open question in biology is how brain processing allows animals, from insects to mammals, to use sensory cues differently and with flexibility according to specific contexts [1]. The navigation of insects provides an ideal system in which to explore this problem, because they face the need to constantly update their memory to forage to new food sources and face new dangers/obstacles on their way [2]. In this context, the substantial use made by central-place foraging insects of specific visual landmarks to retrieve food sources or their nest [3–9] means that efficient discrimination and the ability to learn to direct their paths relative to these landmarks is crucial.

To focus the problem, we consider the different effects that a single landmark cue can have on an ant's behaviour while learning to navigate to a food source. Naïve ants [7], like flies [10] or locusts [11, 12], express a strong innate attraction to a vertical bar, assimilated as a singular landmark; but when trained to find a feeder displaced from this landmark, the landmark can then be used as a reference to keep a constant relative heading and thus return to the feeder location [13]. This simple experimental paradigm already poses several key questions.

The first is how the insect brain controls the maintenance of a specific heading course [14]. Recently, the principle of encoding of orientation in insects has been uncovered following the discovery of a subset of cells (EPGs) in the central complex (CX) functioning as an internal compass [15, 16] similar to mammalian head direction cells [17]. These cells have also been shown to be essential for insects to keep an arbitrary constant direction relative to a single cue [18], a behaviour known as menotaxis [14, 19], suggesting a key role in orientation behaviours in general [20, 21]. However, the neural architecture that allows the encoding of a desired heading and its comparison with an internal compass is still to be determined [18].

The second question is how innate and learned behaviour can interact [7], or more generally, how various parallel pathways controlling directed behaviour could be combined. Recently, Sun et al. [22] proposed a way to combine several navigation systems using a common architecture similar to the ring attractor formed by EPG neurons. The simple summation of these orientation signals could be done directly and offered Bayesian optimal combination characteristics [23] that could support the robustness of insect behaviour [24].

However, a third issue arises from the fact that not all navigation systems will necessarily provide information in the form of a desired orientation. For example, computational models of memory encoding in the mushroom bodies (MB) generally assume the output takes the form of a goodness/badness value of an olfactory cue [25] or a familiarity value of visual scenery [26]. Therefore, the integration of these signals into an ongoing orientation behaviour requires an alternative explanation.

Here we propose an integrative and anatomically constrained model of the insect compass, following the model described by Pisokas et al. [27], combined with a steering mechanism inspired by the model previously proposed by Stone et al. [21], to drive both innate attraction and innate, and learned, menotaxis. With minor, biologically plausible adaptations (see Fig 1), our model supports the formation of a memory based on different sensory inputs. This

**Brain structures/neurons and abbreviations**

| | |
|---|---|
| CX | Central complex |
| EB | Ellipsoid Body |
| PB | Protocerebral Bridge |
| EPG | Ellipsoid body - Protocerebral bridge / Gall |
| PEG | Protocerebral bridge - Ellipsoid body / Gall |
| PEN | Protocerebral bridge - Ellipsoid body / Noduli |
| PFL3 | CX neuron type |
| FBn | Fan-shaped body neurons |
| MB | Mushroom bodies (bilateral brain structures) |
| vPN | Visual Projection Neurons (visual inputs to the MB) |
| KC | Kenyon cell (MB multisensory neurons) |
| MBON | Mushroom body output neuron |
| Vin | Visual innate attraction pathway |

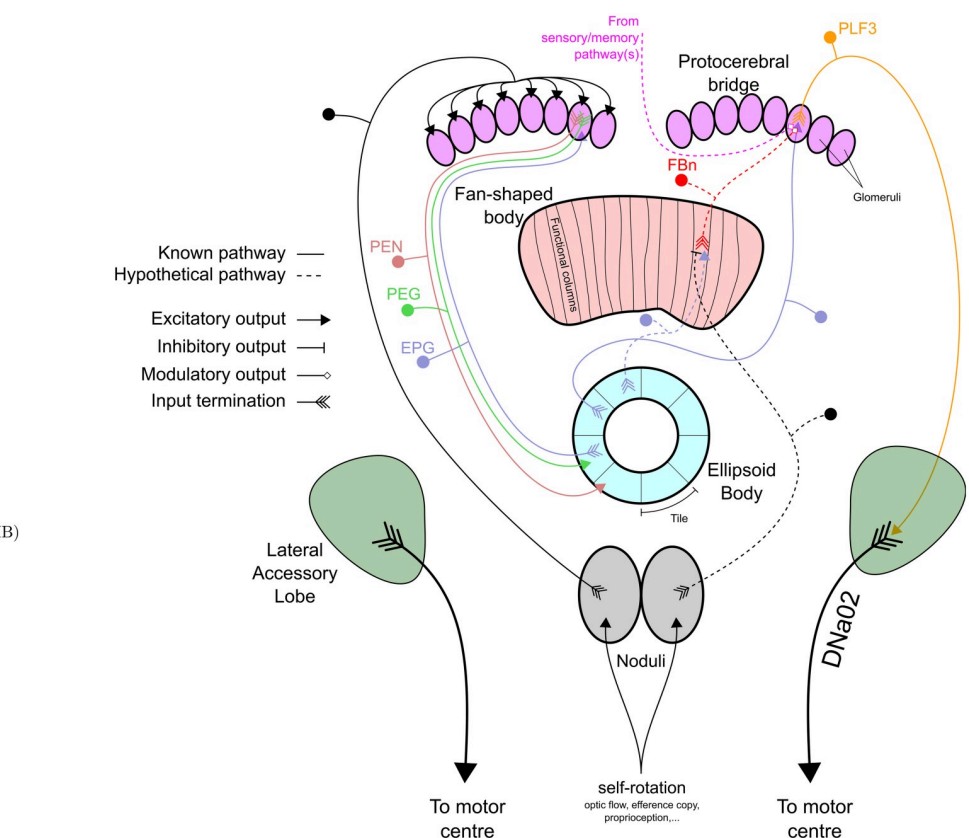

**Fig 1. Abbreviation of brain structures and neurons, and schematic of the CX model.** Indication of the different neuronal pathways used in the model, each pathway is represented by only 1 neuron example showing input and output(s) termination. The left part of the diagram shows the EB compass pathway, the right part shows the steering control pathway. Known pathways are indicated in plain lines while hypothetical pathways are shown with dotted lines.

memory presents some important spatio-temporal properties that match nicely different aspects of insect navigation. First, it matches well with the persistent behaviour observed in *Drosophila* when the visual cues disappear [28]. Then, the scheme of the circuitry allows the integration of external inputs carrying only an information of goodness/badness to produce an appropriate steering, using exploratory behaviour (operant learning). This suggests a generalised mechanism for the different pathways involved in the overall control of movement by the CX. In addition, it permits the inclusion of the reliability which could be helpful to apprehend multisensory integration. This architecture could therefore underlie the integration of both external and internal (such as proprioception or motivational state) sensory inputs in the general orientation scheme of the insect CX. Finally, we demonstrate the capability of the model to describe biological evidence in MB lesions supporting the combined role of the bilateral MB structures.

## 2 Methods and results

### 2.1 Model overview

We start with a pre-existing model of the compass system in the insect brain [27]. This circuit in the CX (specifically in the EB) is composed of three neuron subsets called EPGs, PEGs and PENs (section 2.4) constituting a so-called ring attractor [16]. We assume here a visual input to the EPG neurons, and proprioceptive input to PENs, which creates an activity 'bump' that

tracks the animal's heading [15, 29]. We extend this model to control steering by introducing two sets of PFL3 neurons [30] (one controlling each turning direction, left and right) that have heterogeneous connection strengths from the EPG (section 2.5); this sets a steering direction relative to the bump. We then suggest a mechanism by which these heterogeneous weights can be adapted during behaviour (section 2.6), effectively associating self-motion in a certain direction to a positive (or negative) reward (operant conditioning). This allows steering in a goal direction relative to the stimulus for any arbitrary offset of the bump relative to the stimulus. Finally, we consider how a MB output, signalling familiarity of a visual panorama, could be used as the reward input to the CX and hence establish steering in relation to a route or a visual homing memory (section 2.7). We show this model can account for ablation experiments [13] in which the loss of one MB results in ants exhibiting innate instead of learned attraction (section 2.8).

## 2.2 Simulations

The model was implemented in python 3.6. Each neuron is represented by a simple firing rate model [31]. The activation rule is either a linear function of the input (linear simple perceptron) capped between 0 and 1/-1 (excitatory/inhibitory) or a logic function (active or inactive based on a threshold). The input function is the sum of the activity rate of the pre-synaptic neurons.

The model simulations are conducted in a simple 3D environment containing a single black object in a white background world, comparable with an laboratory experimental paradigm used frequently in different insects and specifically in the experiments we seek to replicate [13]. The 3D environment is a custom implementation in python 3.6 using the pyopenGL library. The agent is placed at the beginning of the simulation at the center of the virtual environment and moves freely until it either reaches the border of a circular area of 100 length unit (*l.u.*) radius or exceeds 5000 time units (steps). The agent has a constant speed of 0.25 *l.u./step* and moves in a straight line following the heading orientation updated every step. During each simulation, a single object is placed randomly in the environment at 150 *l.u.* from its center, thus beyond the border of the agent's arena. This single object, also referred as a cue or landmark, is randomly chosen between a vertical cylinder, a cube or an inverted cone during every individual simulation run. An innate exploratory behaviour is obtained by the addition of a gaussian noise (*sigma* = 10°) in the model steering behaviour, providing a basal exploratory behaviour independent of the CX model output.

## 2.3 Visual circuit

The first layer of the model is a simple visual processing stage. The visual processing is not based on biology, but just provides a simple edge detection process to detect the single landmark used in the simulations. Vision is considered fully panoramic around the azimuthal dimension and from 0° to 80° in elevation (horizon line to the top, note the simulated world contains no information below the horizon). The visual system is segmented into 1296 visual units [48 × 27, inherited from the simulation resolution and the division of the azimuth (48 units) into 8 quadrants (6 units each)], homogeneously distributed on both dimensions. Each unit acts as an edge detector (Fig 2B). For each unit, the summed intensity of the left ($S_L$) and right ($S_R$) half of its visual field is compared (the difference is divided by the sum for normalization) to obtain an index ranging from 0 (no edge) to 1 (vertical edge). The same process is used to compare the top ($S_T$) and bottom ($S_B$) half of the visual unit. Then, the two indexes for the vertical and horizontal edge are averaged to obtain the final output of the unit, a

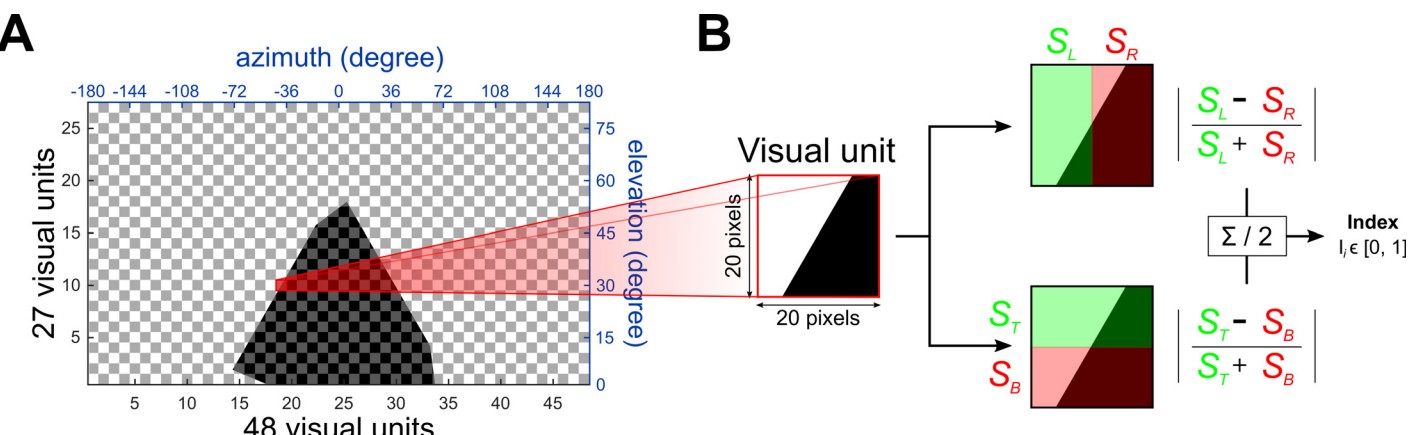

**Fig 2. Model visual system. A.** Map of the 1296 (48x27) elementary units composing the visual system. The visual span is fully panoramic (360˚) on the azimuth and 81˚ upward on the elevation. The red framed visual unit is the one used as example in **B**. **B.** Elementary visual unit process. Each unit acts as a orientation-free edge detector. The comparison of the summed intensity of the left *vs* the right of the unit and of the top *vs* the bottom gives an index from 0 (no difference ↔ no edge) to 1 (corner edge) for each unit.

orientation-free edge index, ranging from 0 to 1:

$$Iv_i = \frac{|S_{left} - S_{right}|}{S_{left} + S_{right}}$$

$$Ih_i = \frac{|S_{top} - S_{bottom}|}{S_{top} + S_{bottom}} \tag{1}$$

$$I_i = \frac{Iv_i + Ih_i}{2}$$

Where $S_{left}$, $S_{right}$, $S_{top}$ and $S_{bottom}$ are the 4 halves summed light intensity values, $Iv_i$ and $Ih_i$ are respectively the vertical and hoizontal edge indexes, and $I_i$ the combined edge index for that unit. $I_i$ is close to 1 if the unit faces a corner ($I_i = 1$ if only a full corner of the visual unit is obstructed), close to 0.75 if the unit faces only a vertical or a horizontal edge, and 0 if it does not face any edge.

## 2.4 Compass: Ellipsoid body model

The main connectivity pattern in the Ellipsoid Body (EB) model is taken from Pisokas et al. [27], who modeled the ring attractor function of the CX that has been highlighted in recent neurophysiological studies in insects [15, 32, 33]. Four neuron types interact to create a ring attractor, three forming the excitatory part of the circuit, EPGs, PEGs and PENs, and one contributing inhibition, Delta7 [27]. The connective scheme between all these neuron types are organised in set of columns in two main parts of the CX, the EB and the Protocerebral Bridge (PB) but functionally form a ring [15, 27], as shown in Fig 3A. The functionality that results from this ring attractor circuit is the creation of a bump of activity in the EB that follow the rotation of the visual environment, creating an allocentric representation of the insects' orientation similar to a compass.

The visual input described above is segmented into 8 equal regions around the azimuth, which (in our model) connect in a retinotopic pattern with the first neuron type, the EPGs (Fig 3B). This retinotopic mapping has been made to simplify in the model the compass function derived directly from a single conspicuous bar, and allow orientation tracking in *Drosophila* during a bar fixation paradigm [15, 34]. Note that in fact, during bar following experiments, the bump of activity in the EB expresses a constant, arbitrary offset with the bar's

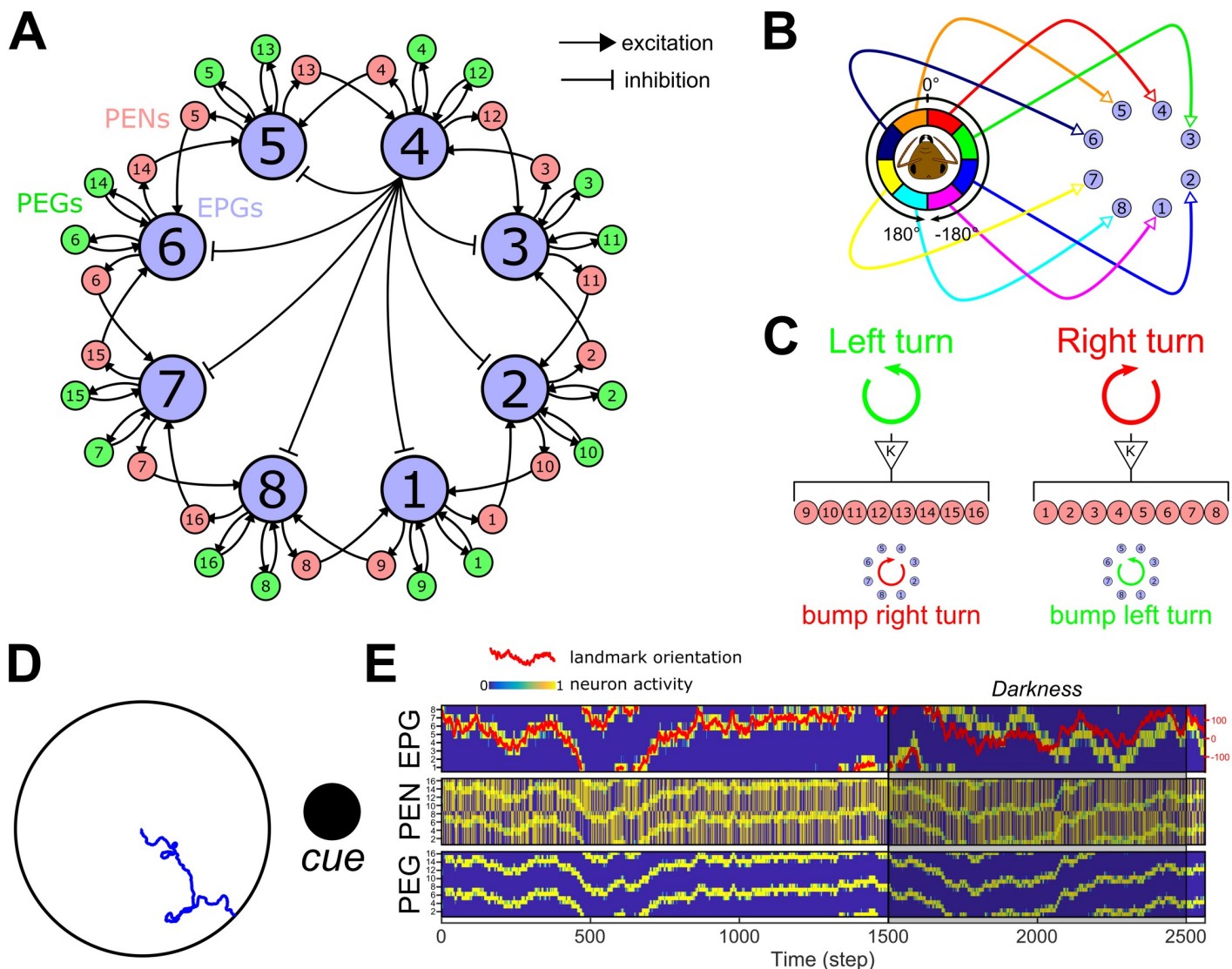

**Fig 3. Ellipsoid body compass model. A.** Connectivity diagram between the main neurons of the EB-PB model, EPGs, PEGs and PENs. EPGs have inhibitory connections with each other as indicated, via Delta7 neurons (black lines; one example is shown from the EPG$_4$ in the central part of the diagram). EPGs (light blue) form a recurrent circuit with PEGs (green) while forming connection with neighbouring EPGs through PENs (pink) on each side. **B.** Connection diagram from the visual circuit to the EPGs. Each angular segment connects in a retinotopic manner to the EPGs. Intrinsically this results in an activity bump forming in the circuit that corresponds to the direction of highest visual contrast. **C.** Proprioceptive inputs to the PENs. When the agent is engaged in a left turn, PEN$_{1-8}$ are stimulated, while during a right turn PEN$_{9-16}$ are stimulated. This will trigger a counter-motion of the bump (posterior view) so that it still indicates the relative direction of the external cue(s). **D.** Example of the function of the EB model in an arena surrounded by a single cue (here a cylinder represented by the black circle). The blue line shows the path of the agent without any influence of the CX model, i.e. only produced by the steering noise. **E.** Activity rate of the three neuron types constituting the EB model. On the first line, the activity of EPGs (blue for no activity to yellow for active) shows a perfect following of the cue orientation in the agent's visual field (red line, scale on the right). Even during a darkness episode (from $1500^{th}$ to $2500^{th}$ steps), the model keeps track of the cue position with a relatively low error thanks to the PENs (second line) and PEGs (third line) combined actions.

real position [15]. The real circuitry from vision to the EB in insects' brain is known to be more complex and involves the Ring neurons, which we will return to in the Discussion. We will also show that the steering mechanism we propose is not dependent on the retinotopic mapping but works for any arbitrary offset (section 2.6; Fig C in S1 Text).

The inhibitory connections between EPG neurons are considered global, as described in studies of *Drosophila* brains [27]. The Delta7 inhibitory circuit is represented as a direct

inhibition from each EPG to all the other EPGs. This creates a winner-take-all circuit, resulting in a single bump in neural activity that will follow the main contrast of the visual scenery. In the simulations here, this is a single object present in the otherwise homogeneous surroundings of the agent.

Inside the same column, EPGs and PEGs form a recurrent circuit, which allows the bump to persist in the absence of visual input (Fig 3E). EPGs also give presynaptic excitation to PENs from the same column, but PENs form presynaptic connections with EPGs from a neighbouring column. In addition, we assume the PENs receive input from the proprioceptive sensory system (self-motion) through the Nodulus [15, 16]. The rotational speed of the agent is measured and injected as a binary input, 0 or 1 modulated by a fixed gain (K = 0.75), to all the PENs on one side of the PB ($PEN_{1-8}$ for a right turn or $PEN_{9-16}$ for a left turn, Fig 3C). This causes the bump to move rightwards or leftwards around the ring, maintaining coherence of the bump position to the external cues even in darkness, as observed in the *Drosophila* brain [15]. The activity rate of each neuron group of the compass model is set by the following set of equations:

$$\begin{cases} EPG_i(t) &= VU_i(t) + 1.0\ PEG_{i|i+8}(t-1) + 2.5\ PEN_{i-1|i+9}(t-1) \\ &\quad -0.2\ \sum EPG_j(t) \\ PEG_{i|i+8}(t) &= 1.0\ EPG_i(t-1) \\ PEN_i(t) &= 0.75\ EPG_i(t-1) + 0.75\ Nod_L(t) \\ PEN_{i+8}(t) &= 0.75\ EPG_i(t-1) + 0.75\ Nod_R(t) \end{cases} \qquad (2)$$

Where $VU_i$ is the input from an octant of visual units (each octant connecting to a single EPG, see Fig 3B), $EPG_j$ the vector of EPGs, excluding the current $EPG_i$, acting through the inhibitory pathway (Delta7), and $Nod_L(t)$ and $Nod_R(t)$ the rotational motion input from the noduli encoding left turns ($Nod_L(t) = 1$) and right turns ($Nod_R(t) = 1$) respectively. The different weights used to modulate the influence of each neuron type have been set to optimize the tracking of the heading even in darkness (Fig 3E).

## 2.5 Steering: EPGs to PFL3s heterogeneous connection scheme

A neurophysiological bridge from the compass to the control of motor behaviour has recently been identified in *Drosophila* [30]. This consists of a subset of neurons, the PFL3s, that are downstream to the EPGs, and upstream to the DNa2 neurons, which are bilateral descending neurons responsible for motor control and steering maneuvers [30]. Moreover, the PFL3s are reported to represent the second largest input to these DNa02 neurons [37], supporting their important role in steering control. We therefore modelled the PFL3 layer to transform the compass bump into a steering command. We used *Drosophila* databases [35–38] to elaborate the connection scheme from EPG to PFL3 neurons (Fig 4), forming synapses at the PB level. This approach has already been used by Rayshubskyi et al. to model the steering behaviour of *Drosophila* [30]. Each brain hemisphere possess 7 PFL3s, which each form synapses mainly with a single EPG wedge (Fig 4A). We assumed that the DNa2 of each hemisphere integrate the inputs from the whole set of PFL3s of the ipsilateral hemisphere, as each DNa02 has been shown to be acting independently [30]. Thus, the influence of each EPG depends only on its synaptic connection to one or more PFL3 on each side. We therefore represent two PFL3s (one on each side) for each EPG (Fig 5A), and set the synaptic weight between them to the sum of synapses from each EPG tile (Fig 4B) to a PFL3. Note this means a PFL3 cell in the

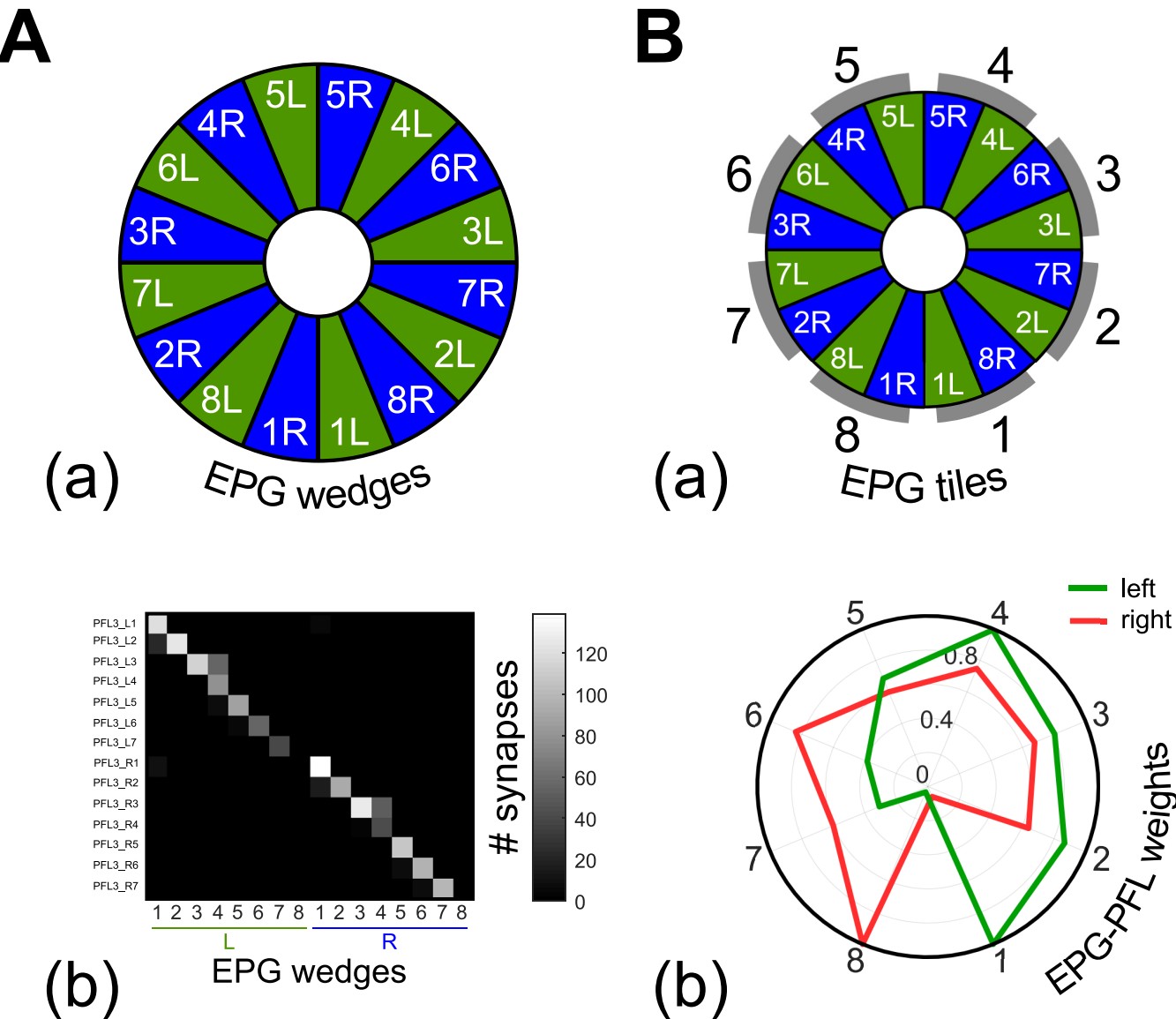

**Fig 4. The EPG to PFL3 connectome at the interface between the compass and the steering control.** The full connectome is assessed from online *Drosophila* brain databases (https://v2.virtualflybrain.org, [35–38]). The full list of the synaptic connection between neurons annoted PFL3 and EPGs is given in S1 Table. **A.** (a) Division of the EB in wedges 15. (b) Quantity of synapses expressed as a function of the EB wedges and the PFL3 columns. **B.** (a) Relationship between EB wedges and the EPGs modeled here (Fig 3A). (b) The connectome is transformed into EPG-PFL3 synapse weights to create the steering model (see section 2.5). Weights are obtained by summing for each EPG the synapses on the right or on the left part of the PB. The gains are then normalized to fit between 0 and 1.

model could represent from 0 to 3 actual PFL3 cells, with a corresponding weights from zero to one which we obtain from the *Drosophila* database [35–38](i.e., the weights are normalized). The synapses between EPGs and PFL3s are represented as excitatory (Fig 5A) as EPGs are reported to form cholinergic synapses with PFL3 [35–39]. The output of the PFL3 neurons are determined by the following equation:

$$PFL3_i(t) = EPG_i(t-1) \; \omega_i^{EPG-PFL3} \tag{3}$$

With $\omega_i^{EPG-PFL3}$ the synapse strength between an EPG neuron and the corresponding PFL3

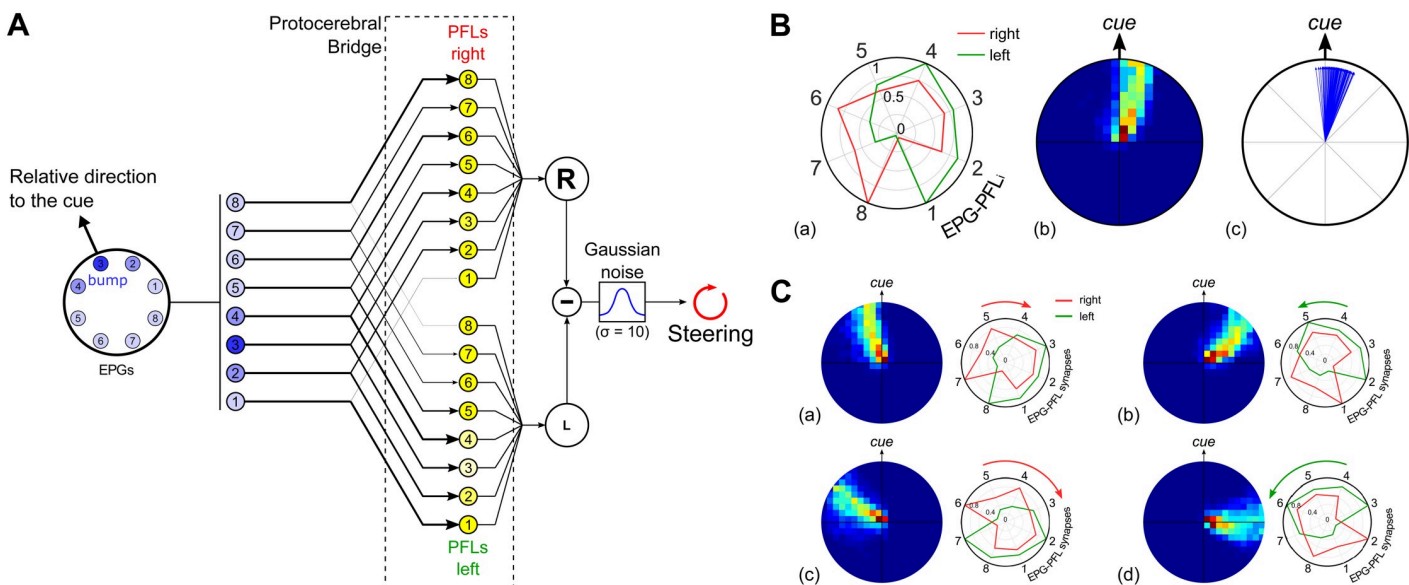

**Fig 5. EPG to PFL3 connectivity scheme. A.** Diagram of the EPGs to PFL3s connectivity that drives the steering. Each EPG makes inhibitory connection to two PFL3 neurons, one on each side. The strength of the synapse depends on the index of the EPG$_i$ as shown in B. The summed activity rates of the right and left PFL3 neurons are compared to obtain the CX steering signal. Finally, gaussian noise ($\nu = 0$; $\sigma = 10$) is added to obtain the final steering command. **B.** (a) Synapse strength as a function of the index of EPG$_i$, obtained from the connectomic approach (Fig 4B; [30]). Synapses are inhibitory and therefore multiplied by -1. Red lines represent the connection with the subset of PFL3s from the right part of the PB and green from the left part. Synapse weights are presented in a circular representation to show the correspondence with the EPG input geometry. Index 1 corresponds to the synapse between ellipsoid body tile EPG$_1$, connected to the rear right FOV (-157.5°), and PFL3$_1$, and so on, through to index 8, the synapse between EPG$_8$, connected to the rear left (157.5°), and PFL3$_8$. (b) Heatmap in the arena from 100 simulations in total. The data are normalized to the landmark direction (indicated on the right). (c) Final direction/position of each simulation (n = 100) with the cue placed in different positions around the arena, expressed in the cue reference frames. **C.** (a) Simulations (n = 100 for each shift condition) where EPG-PFL3 synapse weights were shifted for 1 cell to the right/clockwise (compared to **B.**). Results show the heatmap relative to the cue (top) of the 100 simulations. (b) Simulations (n = 100 for each shift condition) where EPG-PFL3 synapse weights were shifted for 1 cell to the left/anti-clockwise. Results show the heatmap relative to the cue (top) of the 100 simulations. (c)-(d) Equivalent to (a) and (b), respectively, with a 2 cells shift.

neuron. The unilateral sum of the PFL3 outputs is therefore considered to represent the motor signal received by each DNa02 neuron [30] and their comparison directly gives a steering command, calculated as follows:

$$\Delta_{steer}(t) \quad = \quad \sum PFL3_i^{right}(t) - \sum PFL3_i^{left}(t) + \epsilon_{steer}^{10} \tag{4}$$

With $\Delta_{steer}$ the change in direction taken by the agent at time t, and $\epsilon_{steer}^{10}$ the gaussian steering noise with a standard deviation of 10°. The steering signal produced by the PFL3s populations comparison is constrained to a [-2.5 2.5]°.step$^{-1}$ range to avoid unrealistic turning speed and instability. Note that this constraint on the model output is smaller than the steering noise applied and, therefore, preserves stochastic exploratory behaviour (see next section 2.6) while guiding the general direction of motion. Please note that we did not consider here the polarity (excitation/inhibition) of the PFL3 to DNa2 synapses but rather use the creation of a left-right difference to drive the steering. However, it has been reported PFL3 neurons from the left (respectively right) PB connect directly onto DNa02 on the right (left) [40]. In addition, considering a greater activation on the right DNa02 (compared with the left) produces a leftward turn [30], our PFL3 synapses to the DNa02 can be considered inhibitory (left activation promote left turns), as it has been assumed by Hulse et al. [40] whereas Rayshubskiy et al. [30] assumed the opposite.

To generate a steering signal in the PFL3 layer, we propose here that the observed heterogeneity in the distribution of EPG-PFL3 synapse weights is ideally suited to give rise to a Left-Right inequality and thus to a turn. More specifically, the specific weights/connections pattern

(Fig 5B) that frames the EB bump to the front of the visual field would suffice to generate an innate attraction to the conspicuous landmark. The detection of the cue in the right part of the visual field (as schematised in Fig 5A) would lead to an unbalanced excitation of the PFL3s and thus a increased signal to turn right and ultimately centering the cue in the frontal field of view (equally the detection of the cue in the left part of the visual field would have the opposite effect). The general direction taken depends upon the intersection between the left and right EPG-PFL3 synaptic weights, the steering noise only creating some stochasticity in the behaviour and a convoluted path as a consequence. When tested in the virtual environment, mimicking indoor arena experiments with a single landmark classically used in wood ants [13, 41], we observed accordingly an innate attraction behaviour toward the cue (Fig 5B).

Crucially, assuming heterogeneous weights (or connections) occur between EPG and PFL3 neurons provides the flexibility to consider the adaptative outcome if the weight distribution is altered. For example, it could be shifted to the right or the left (Fig 5C) to maintain the bump in a different position other than the front of the agent. The subsequent oriented behaviour is opposite in direction to the shift of the EPG-PFL3 synapse weights, because they define the position in which the bump will be maintained during behaviour. Thus, by setting the weights accordingly, the model is capable of producing either direct attraction to a cue or menotaxis at any offset from the cue direction. We now go on to explore how the weights, instead of being pre-determined, could be plastically altered through the agent's own experience to produce appropriately a flexibility of the oriented behaviour in different contexts.

## 2.6 A memory to modulate the EPG-PFL3 connection weights

In order to allow insects to deal with different goals at different times, the mechanism that orients them with respect to landmarks, visual memories or other external information should allow some kind of plasticity or adjustment over time. Here we propose a mechanism that adopts some characteristics of the model proposed by Stone et al. for path integration [21], based on neuroanatomical evidence in bees, that accumulates a homing vector from the optic flow speed estimation and the sky compass. Specifically, we add a set of units (similar to the CPU4 units in [21] model) that get both compass and self-motion input, and implement a one column (left or right) shift [42] before connecting to the steering neurons (PFL3 in our model—CPU1a in [21] model). In the present model we limited the self-motion input to the perception of rotations (in a bi-modal, left or right, way), which is used to inhibit ipsilaterally the update of the synaptic weights between the EPG and PFL3 neurons, rather than directly contributing to steering (Fig 6). The existence of such input, potentially provided by the noduli, could be supported by recent observation in *Drosophila* at the PB level [42]. Stone et al. [21] proposed that CPU4 neurons (PF-LC in *Drosophila*), arborizing between the FB and the PB, could encode the memory for the path integrator. In contrast, here we propose that the modulation that would drive the use of landmarks could be represented rather by a direct plasticity between EPGs and PFL3s. Several studies have linked the FB to the encoding of desired direction in insects brain [43–45]. In addition, PFL3s receive upstream inputs from synapses in the FB [38], we therefore consider it as a target for potential modulatory neurons and refer to the neurons in our model as FB neurons (FBn). In addition, we set the formation of this memory under the control of a global reward signal representing the sensory pathway(s) influenced by the desired heading [18]. The aim of the weight modulation is to enable the formation of a similar spatial weight pattern as presented in the previous section. We specifically decided to apply the memory by the mean of a modification of the synaptic weight here for consistency with the connectomic approach presented in the previous section (2.5). However, we recognize that there is no evidence so far of EPG-PFL3 synapse plasticity and that the

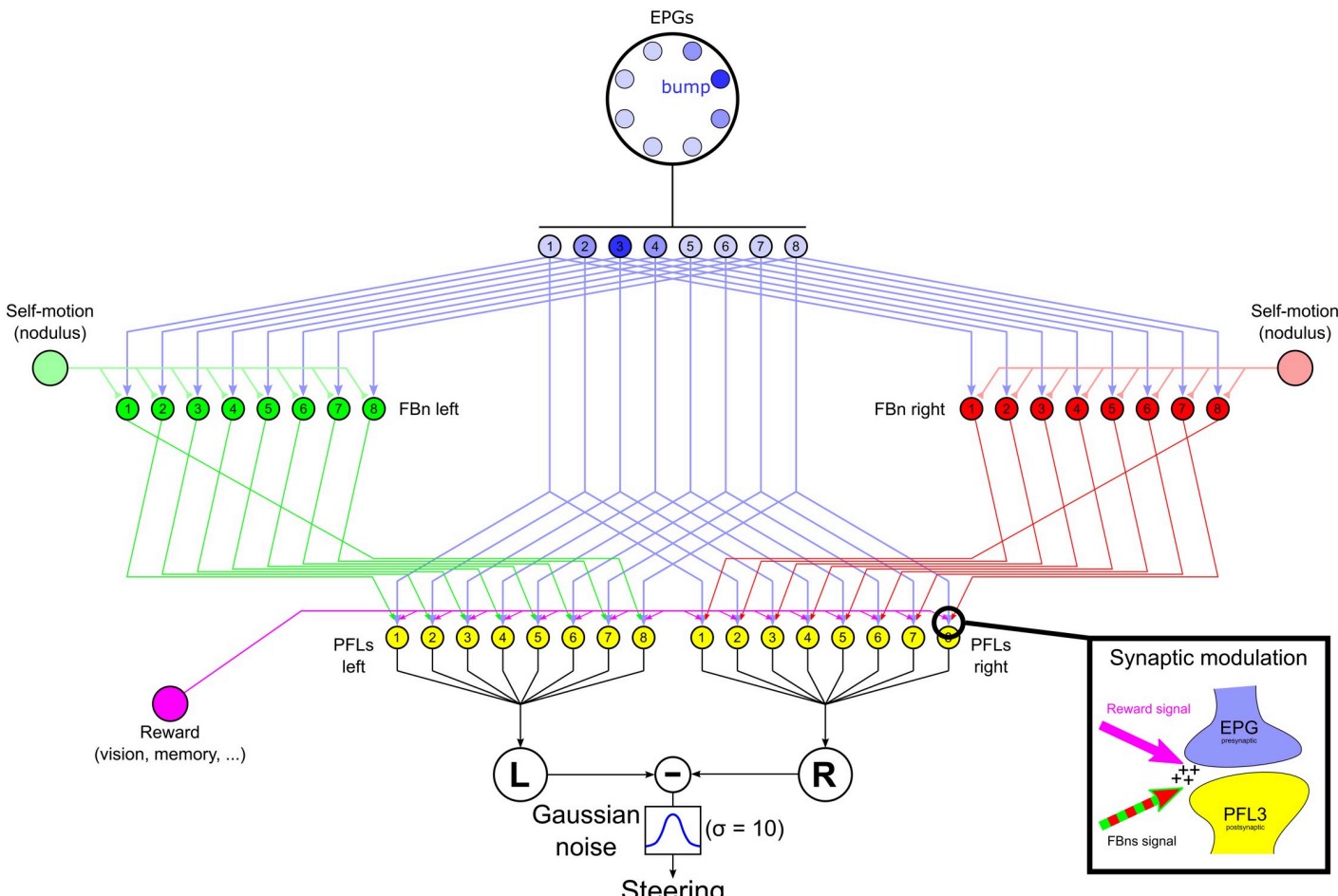

**Fig 6. Steering CX model with external control.** The central part of the diagram is similar to the model of EPG to PFL3 connection presented in Fig 5A except the synapse weights are initially set homogeneously ($\omega_i^{EPG-PFL3} = 0.5$). On each side the model copies the [21] model architecture such that the connection between FBns [red (right FBns) and green (left FBns) cells] and steering neurons (PFL3s, yellow cells) are shifted by one column, in opposite directions on each side. FBns also receive rotatory self-motion information inhibiting inputs [light red (right turns) and green (left turns) lines] for the ipsilateral side, therefore the right FBns are activated during left turns and viceversa. The reward signal (pink) is provided by the sensory pathway indicating goodness-badness of the current sensory input. The modulation of the EPG-PLF3 synapses is under the control of this reward signal and the FBns as shown in the bottom-right inset. The steering command is obtained by comparing the right and left summed activity of the PFL3s after the addition of a gaussian noise ($\nu = 0; \sigma = 10$) as previously (Fig 5A).

memory could, for example, be accumulated at the FB level, as it is proposed by a CPU4 accumulation in Stone et al. PI model [21]. In fact, for the level of abstraction used here, such an implementation would be computationally equivalent (as demonstrated in Fig C in S1 Text). We rather focus on the potential of the functional columnar shift [21] and of the convergence of the three following components: the self-motion perception, the compass information and the influence of goodness-badness encoded sensory pathways. The FBns activity is set by the following equation:

$$
\begin{cases}
FBn_i^R(t) &= EPG_i(t) - NodR(t) \\
FBn_i^L(t) &= EPG_i(t) - NodL(t)
\end{cases}
\tag{5}
$$

With *Nod* the inhibitory self-motion signal, from right ($NodR(t) = 1$) and left ($NodL(t) = 1$) turns respectively. Therefore $FBn_{1-8}^R(t) = 0$ during right turns and $FBn_{1-8}^L(t) = 0$ during left

turns. Then, the EPG to PFL3 synapse weights are updated as follows:

$$\begin{cases} \omega R_i^{EPG-PFL3}(t+1) & = & \omega R_i^{EPG-PFL3}(t) + \alpha\, FBn_{i-1}^R(t)\, Rew_{CX}(t) \\ \omega L_i^{EPG-PFL3}(t+1) & = & \omega L_i^{EPG-PFL3}(t) + \alpha\, FBn_{i+1}^L(t)\, Rew_{CX}(t) \end{cases} \tag{6}$$

Where $\omega R/L_i^{EPG-PFL3}$ is the synapse weight between $EPG_i$ and $PFL3_i$ on the right (R) or on the left (L) FB. The synapses are initialized in the model with a value of 0.5 and restricted between 0.2 and 0.8 during the simulations. $\alpha$ is a free parameter set equal to 0.001. The reward signal ($Rew(t)$) is set (as we describe below) to act as modulator for the synapse update by the FBns of the neighbouring functional column on the right (for the left PFL3s) or the left (for the right PFL3s). This learning rule is used as an efficient way to implement the key conceptual function, that the compass shift associated with rewarded actions should bias the steering weights, rather than being intended as a biologically justified plasticity mechanism; as already discussed, and shown Fig C in S1 Text, there are alternative mechanisms that could serve the same purpose.

The effect of Eqs 5 and 6 is as follows. Say the bump is centred on $EPG_3$, as shown in Fig 6, and the agent is turning right and observes a view which elicits a reward above threshold from any sensory system. The right FBns being inhibited by the right turn, the left FBns will induce an increase, under the control of the high reward level, in the EPG-PFL3 synapse shifted by one column to the left (counter-clockwise) from the EPG bump, on the $EPG_4$. This leads to the increase of the tendency to turn right if the cue was to appear on the left of its current rewarded orientation. It therefore induces a memory of the motor action that led to a rewarded orientation, and thus corresponds perfectly to an operant behaviour. By exploring the environment, this mechanism creates two weight distributions that frame the position of the EPG activity bump at its maximally attractive orientation in the local environment.

To reproduce the innate attraction to a simple cue, widely observed in insects [46, 47], we created a reward signal provided by the visual processing used in this paper. The aim is to have reinforcement of the signal from the frontal ommatidias (i.e. when facing the cue). This can be done using a mask on the visual units as shown in (Fig 7, top panels). The mask is multiplied with the visual input and the output summed across all units to provide the reward signal:

$$Rew_{CX}(t) \quad = \quad \sum M_i\, I_i(t) \tag{7}$$

With $Rew_{CX}$ being the CX reward signal used to update the EPG-PFL3 synapses, $I$ the matrix of the edge detection indexes (Eq 1) estimated by the visual units (Fig 2) and $M$ the mask matrix (same size as $I$). We tested alternative masks ($M$) with either a sharp or graded preference (Fig B in S1 Text). for a frontal orientation of the strongest contrasts, and either with or without negative responses for contrasts at the rear of the visual field (Fig 7A and 7B for sharp masks and Fig B in S1 Text for graded ones). The aim of these masks, rewarding contrasts in the frontal part of the visual field, is to reinforce the weights so as to stabilize the landmark in the center of the visual field and therefore produce attraction to it.

Using the architecture presented in Fig 6, we show its capability to generate a memory able to drive an innate attraction behaviour toward a single cue (Fig 7A). As observed with the synaptic weights inherited from the native connectomic (Fig 5B), the weights produced by the model define the general direction taken by the agent at the intersection of the left and right sets of weights. In addition to adding stochasticity to the path, the noise added to the steering behaviour is crucial here to generate the initial exploration necessary for the synaptic memory acquisition. Even using a discrete filter, which does not carry any information about the left or right displacement of the cue from the current heading, the model generates a consistent behaviour and coherent weight distribution for the EPG to PFL3 synapses (Fig 7A). The use of

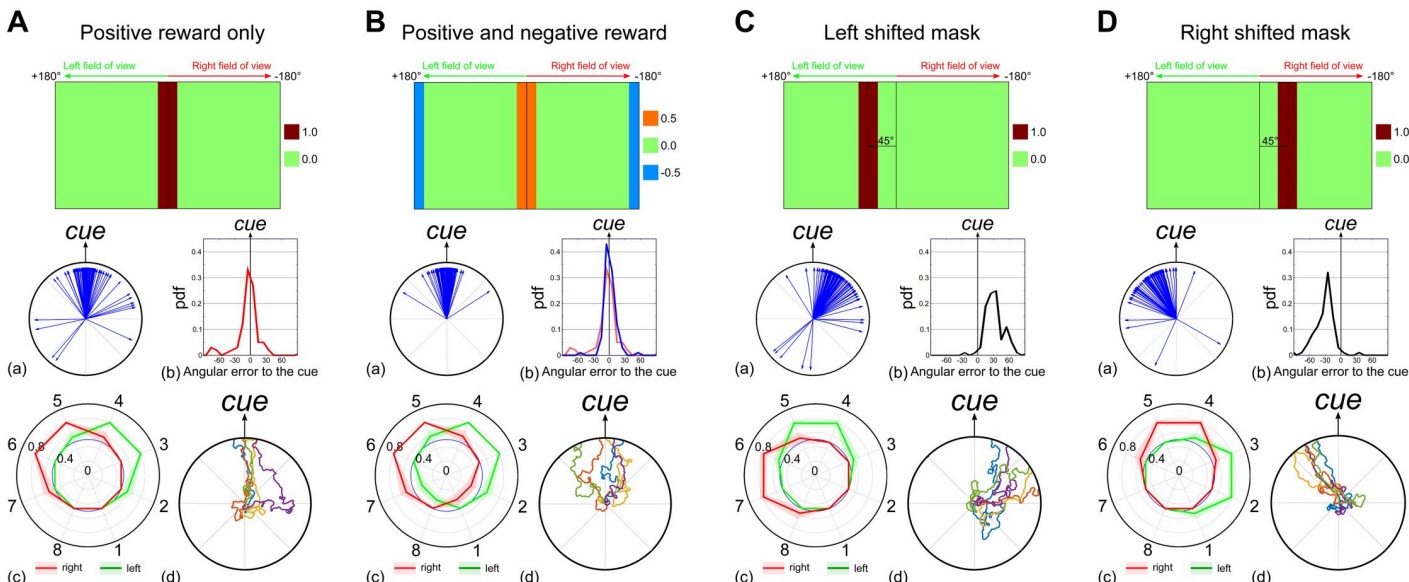

**Fig 7. Innate behaviour under the control of a visual reward signal.** Simulations of the FB steering model (Fig 6) using a reward signal provided by the visual processes to modify the EPG-PFL3 synapse weights. We created the visual reward signal to the CX using different masks. Results for each panel include (a) the final path directions (n = 50 simulations), (b) the probability density function of the final direction vector relative to the cue orientation, (c) the averaged EPG-PFL3s synapse weights and (d) examples of 5 simulation paths. See also Fig B in S1 Text for results with continuous masks. **A.** Visual reward to the FBs is equal to the sum of the visual units signal through a discrete mask equal to 0 outside the 30˚ frontal area or 1 inside. **B.** Visual reward to the FBs is equal to the sum of the visual units signal through a discrete mask equal to 0.5 inside the 30˚ frontal area, -0.5 inside the 30˚ rear area, or 0 otherwise. **C.** The mask is shifted by 45˚ to the left of the visual field. The reward area extends from 30˚ to 60˚. **D.** The mask is shifted by 45˚ to the left of the visual field. The reward area extends from -30˚ to -60˚.

a negative sensory input, when the cue appears at the rear of the visual field, to the model improves the performance (10% more path ending at the cue orientation, Fig 7B). The negative input is more essential when a graded preference function is used, otherwise experience of some positive reward in all directions leads to random preferred orientations (Fig B in S1 Text). In both cases (continuous or discrete masks), the advantage of positive and negative sensory inputs is of interest considering that insects are known to associate (innately or learned) positive or negative values to olfactory [48] and/or visual cues [2]. Furthermore, the modification of the mask, to reward the cue in a shifted position, produces an oppositely shifted behaviour (Fig 7C and 7D), showing the ability to use the sensory information to drive oriented behaviour relative to a landmark, similar to menotaxis [14, 19].

The formation of a memory in the CX also permits a sustained behaviour when directly attractive cues disappear in the environment. To illustrate this we tested the model in static simulations, where the agent is maintained in the center of the arena and can only rotate, and we made the cue disappear after a given time, necessary to form the EPG-PFL3 synaptic memory (Fig 8A). The persistence of the behaviour depends on the capability of the EB compass to maintain its function in darkness [15]. The accumulation of error along time in absence of the landmark produces a gradual shift in the behaviour (Fig 8C.a). Alternatively, if we complement the compass with an absolute reference in the environment, which could be provided in insects by sky cues ([49, 50], Fig 8B), the agent can maintain a consistent straight direction for an indefinite period after the disappearance of the attractive landmark (Fig 8C.b).

Note that for these results (excepting Fig 8C.b) the bump and the landmark which generates the attraction were directly attached, i.e., the ring-attractor compass represents a truly allocentric compass based on the external landmark cue. In contrast, in the insect brain during bar-following paradigms it is known that while the movement of the EPG bump follows the bar movement, the actual position of the bump has an apparently arbitrary offset from the bar

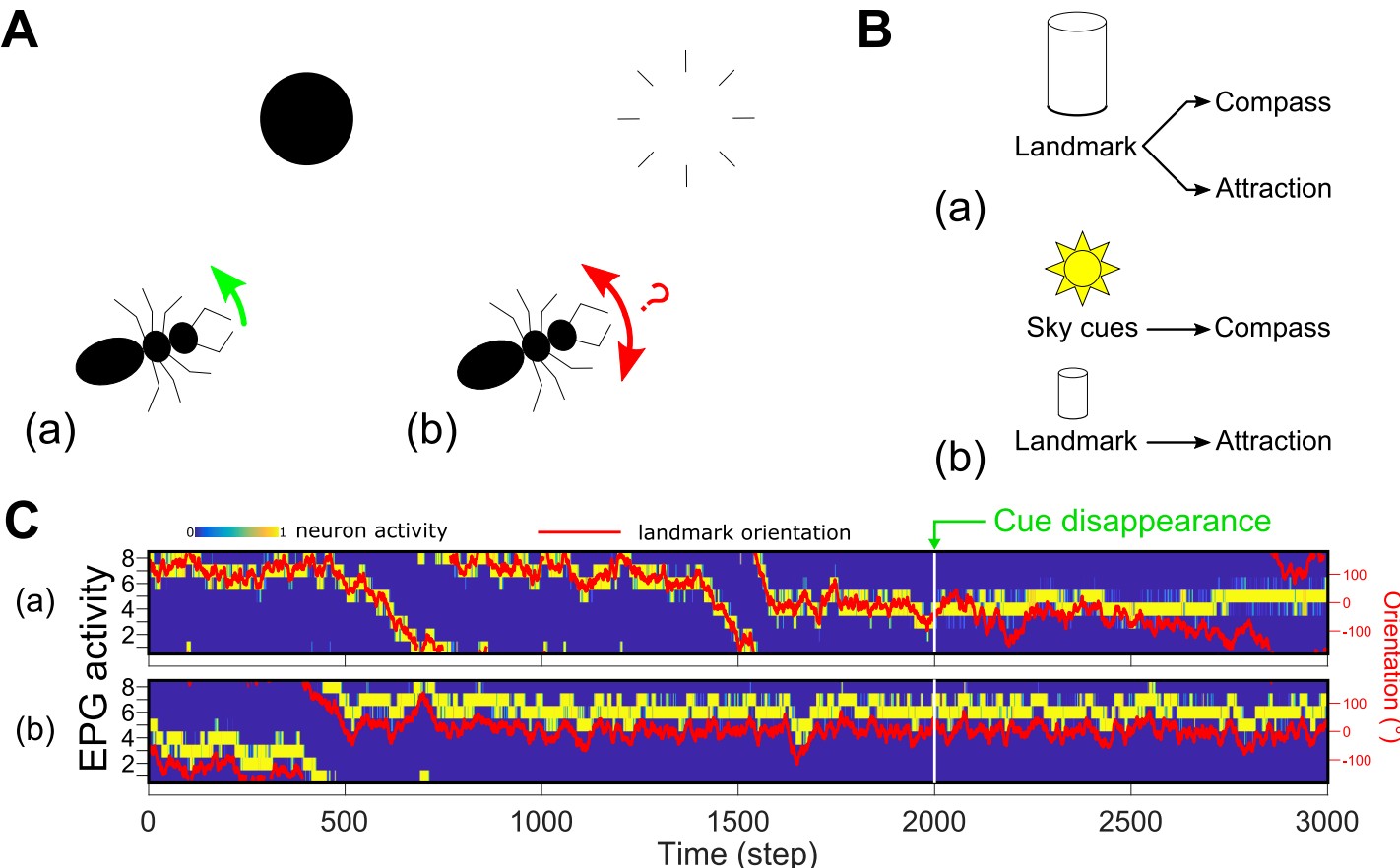

**Fig 8. Persistence of direction in the absence of the landmark. A.** (a) Simulations are done in a static mode, i.e. the agent never leaves the center of the arena and only rotates around its vertical axis. (b) After the 2000$^{th}$ step the cue disappears. **B.** (a) We tested the model in the same configuration as before, where the landmark is used both as the main cue for the compass and to generate the directed behaviour (Fig 7C). Results are shown in **C.**a. (b) We tested another configuration where the compass was provided with cues from the absolute orientation in the environment (such as sun position or sky polarization could provide). The attraction behaviour is kept based on the landmark as before. Results are shown in **C.**b. **C.** EPG activity rate (blue to yellow) and relative landmark orientation (red line, right scale) during simulation of the cue disappearance paradigm showed in **A.** (a) The compass and the attraction are both based on the landmark (**B.**a). The direction can be maintained but slowly drifts as there is no external reference. (b) The compass is based on an absolute orientation perception (potentially from sky cues) while the attraction is based the landmark (**B.**b). Note this creates a potential offset in the position of the bump and the landmark, but the agent still moves towards the landmark. The heading can be accurately maintained when the landmark disappears.

azimuth [15]. To explore this, we reproduced such an offset between the EB bump and the visual cue's position in the visual field by applying a random shift in the connectivity between the visual units and the EPGs (Fig D in S1 Text). This means each simulation expresses a unique offset between the bump and the landmark. However, the innate attraction behaviour to the cue produced by the model is unchanged as the model defines the correct set of weights to drive the behaviour toward the landmark in any situation. Thus, the model does not use the bump as an absolute compass but rather as a reference point to interpret another signal (here the valence signal from a sensory pathway) to define the orientation to take.

## 2.7 Incorporating a visual familiarity memory from the mushroom bodies

In addition to using a single visual input to generate an innate attraction to single object, as described above, we also tested how the output of the Mushroom Bodies (MB), a bilateral structure tightly associated with memory in the insect brain [51], could be used to generate a local memory in the CX to control appropriate orientated behaviour. To do so, we adapted the

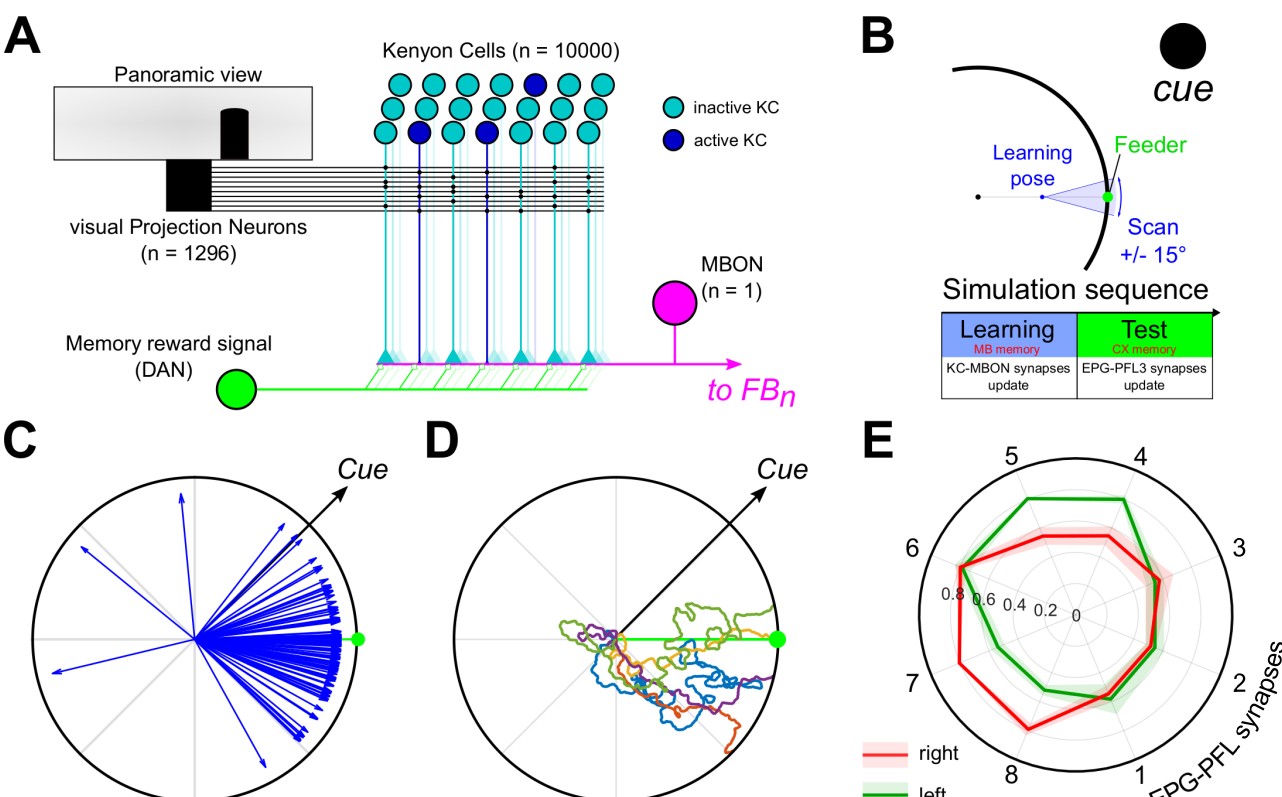

**Fig 9. Orientation to a goal direction supported by the MB long-term visual memory. A.** Mushroom bodies model diagram. The visual Projection Neurons (vPN) correspond to the visual units (Fig 2) extracting edge from the panoramic view. Each KC has a random post-synaptic connection pattern with 2 to 5 vPNs and a pre-synaptic connection to the MB output neuron (MBON). During the learning procedure, the memory reward signal (DAN) is activated and induces the decrease of the synaptic strength between the activated KCs and the MBON. The MBON activity is then used as the reward signal to adapt the CX circuit. **B.** Schematic of the learning phase before the simulation. The agent faces the feeder during 100 steps from a position in between the feeder and the center of the arena (learning pose), with an orientation error from −15 to 15˚. The MB model memory is updated during this phase (DAN active). **C.** Example paths from the retrieval experiments using the MB model as an input for the CX model. **D.** Final direction vectors for 50 simulations with the feeder at 0˚ and the cue oriented at 45˚. The red arc shows the median (dot) 95% C.I. obtained by bootstrapping (rep = 10000). **E.** Averaged right (red) and left (green) EPG-PFL3 synapses weights (shaded area: ±*s.d.*) obtained from 50 simulations.

model of familiarity encoding for route memories that has been hypothesised to occur in the social insects MB [26], and use its output as the reward input in our model. Recently, experimental results have supported the importance of MBs to view-based memory and navigation in ants [13, 52, 53]. The direct influence of a MB memory is also supported by the existence of several MBONs largely projecting into the FB in *Drosophila* [54].

The MB is modeled as a network (Fig 9A) composed of 10000 Kenyon cells (KCs) each sparsely connected to 3 to 5 visual Projection Neurons (vPNs). vPNs are directly taken as the outputs of the visual units calculating the edge indexes from the panoramic view (Fig 2). The long-term memory is formed by a simple associative learning rule at the interface between the KCs and the output of the MB (MBON, Mushroom Body Output Neuron). The MBON signal is calculated as the sum of the KCs activity [$KC_i$—binary, either 0 (active) or 1 (inactive)] weighted by their associated synapse weight ($\omega_i^{KC-MBON}$) and normalized by the number of KCs active, thus maintaining the MBON value between 0 and 1:

$$MBON \quad = \quad \frac{\sum \omega_i^{KC-MBON} KC_i}{\sum KC_i} \qquad (8)$$

**Table 1. Mushroom bodies visual long-term memory learning rules.** KC to MBON synapses are updated depending on the KC activity level and a reward signal coming from dopaminergic neuron (DAN), considered on (1) during the learning phase and off (0) otherwise (during the test phase). Only the combination of an active KC with an active DAN induce the reduction of the corresponding KC-MBON synapse weight.

| DAN(t) | $KC_i(t)$ | $\Delta_i^{KC-MBON}(t+1)$ |
|--------|-----------|---------------------------|
| 0 | 0 | 0 |
| 0 | 1 | 0 |
| 1 | 0 | 0 |
| 1 | 1 | -0.2 |

The KC-MBON synapses weight are initially set as 1 and updated under the control of a rewarding signal (DAN, Dopaminergic Neuron) following the learning rule (inherited from classic MB associative memory models [26, 51]):

$$\omega_i^{KC-MBON}(t) = \omega_i^{KC-MBON}(t-1) + \Delta_i^{KC-MBON} \qquad (9)$$

Where $\omega_i^{KC-MBON}$ ($\in[01]$); and $\Delta_i^{KC-MBON}$ is given in Table 1.

The learning phase is made to mimic learning walks/flight observed in insects during which they face a targeted location (nest, food) and acquire some visual memory [55–57]. Therefore, the DAN activity is kept to 1 during the whole learning phase to generate the visual memory. The agent is placed at the mid-distance between the center of the arena and a virtual feeder (always at 0˚ at the edge of the arena) and scans the environment during 100 time steps (0.3˚/ step) over a range from −15 to 15˚ centred on the feeder (Fig 9B). This range is chosen arbitrarily to match the span of the discrete visual mask (Fig 7A). At the end of the learning phase the DAN activity is set to 0 during the test phase to avoid the development of new MB memories.

Following the MB learning rule, which decreases the synapse strength from KC to MBON during learning episodes, the weaker the signal from the MBON, the stronger the familiarity of the current visual scene. Therefore, the MBON output is translated into a positive reward for greater familiarity, ($Rew_{CX}(t) = 1 - MBON(t)$), and used to influence the EPG to PFL3 weight updates as before. To avoid the influence of the noise at low familiarity, we used a threshold on the MBON value so as any value lower than 0.25 is equal to 0.0.

Using the MB familiarity of the visual scene as the reward input to the CX model enables the agent to be directed towards the feeder location (Fig 9C and 9D). Despite the restriction of the memory to a single episode at a single location, and the noisy familiarity observed during the simulations, the model is able to induce direct oriented behaviour toward the goal location albeit sometimes after expressing an initial erratic behaviour, due to the steering noise. The averaged EPG-PFL3 synapses weights clearly show a pattern to keep the landmark/bump in the correct orientation [≈45˚ on the left (counter-clockwise)] to reach the virtual feeder (Fig 9E).

## 2.8 Lesion experiments

In order to link our model to biological experiments [13], we set up simulations where the MB could be fully silenced (i.e., output equal to 0). The mushroom bodies are present as a pair of structures in insect brain. Therefore, the first step was to duplicate the MB used in previous simulation to obtain two identical structures, one for each hemisphere. Both MB follow the same vPN-KC connectivity parameters (as previously shown, Fig 9A), but their connectivity is independent. In addition, each MB receives homogeneous inputs from the entire set of vPNs

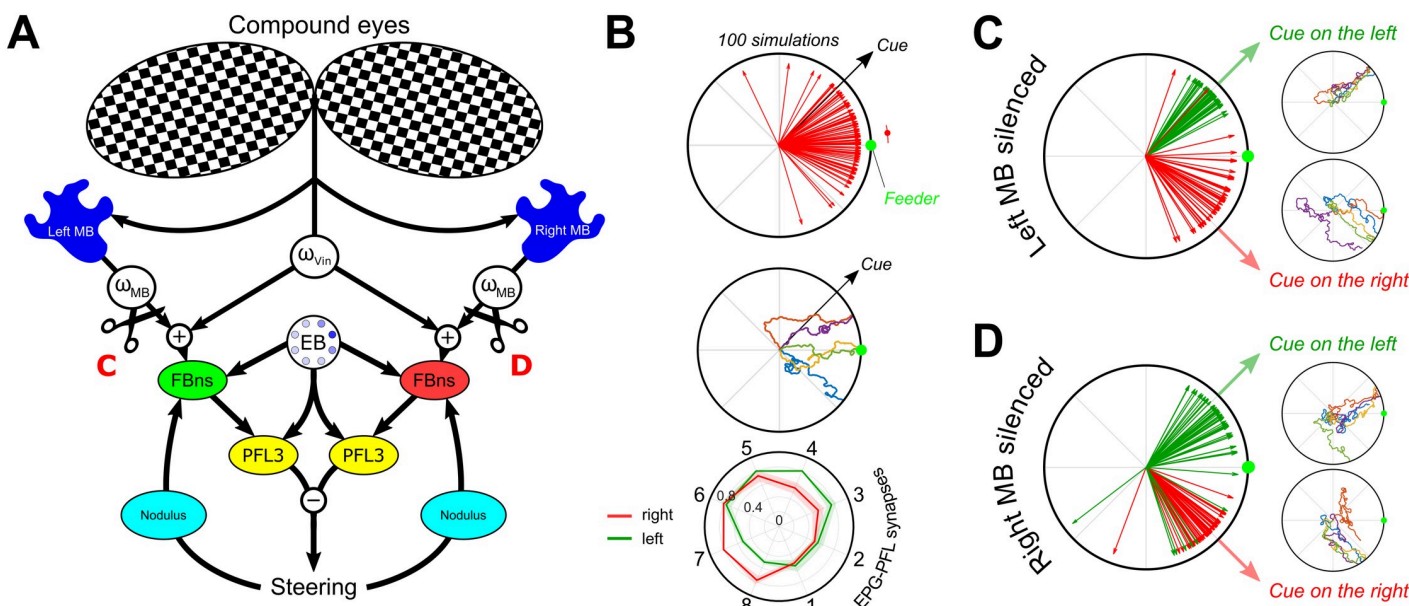

**Fig 10. Replication of unilateral mushroom body lesion experiments from [13]. A.** Schematic of the model including both MB and innate visual attraction. Both elements are linearly summed to form a single reward signal, using individual weights ($\omega_{Vin}$ & $\omega_{MB}$) to modulate the influence of each component. Lesions are indicated by the black scissors and the results are reported in **C** and **D** (Left and right lesion respectively). **B.** Simulation with the combination of innate attraction ($\omega_{Vin}$) and the mushroom bodies ($\omega_{MB}$) pathway. The learning procedure is similar as presented in Fig 9B. From top to bottom, final directions (blue arrows, $n_{sim}$ = 100) taken during the simulations [blue arc indicate the median (dot) 95% C.I. obtained by bootstrap ($n_{rep}$ = 10000)], examples of path obtained during simulations, and averaged right (red) and left (green) EPG-PFL3 synapse weights (shaded area: ±*s.d.*) obtained during 50 simulations. **C.** Unilateral lesion of the left mushroom body. Final direction vectors and example paths. **D.** Unilateral lesion of the right mushroom body. Final direction vectors and example paths.

from the panoramic view of the agent, i.e., the connectivity from both eyes is presupposed identical (Fig 10A). It is not known to what extent the mushroom bodies receive inputs from the contralateral compound eyes, however, anatomical [58] and behavioural [13] observation supports the existence of cross-over in ants. The MB output consists on each side of a single MBON neuron which connects to the FBns of the same side only. Therefore, the ablation of one MB deprives the ipsilateral FBns of any memory related inputs. To keep the model to the simplest case scenario we linearly added both innate visual (*Vin*) and MB inputs to generate a common reward signal to control the memory as follows:

$$Rew_{CX}(t) \quad = \quad \omega_{MB}\, MBON(t) + \omega_{Vin}\, Vin(t) \quad with \begin{cases} \omega_{Vin} & \in & [0.5\ 1.0] \\ \\ \omega_{MB} & \in & [3.5\ 4.5] \end{cases} \quad (10)$$

With *Vin* the signal provided by the visual mask (see Eq 7), $\omega_{Vin}$ the gain modulating the visual innate attraction pathway and $\omega_{MB}$ the gain modulating the MB pathway. For each simulation, a pair of gains is randomly selected from a uniform distribution in the range indicated in Eq 10. The ranges of the pair of gains are set to advantage the MB pathway (approximately with a factor 4 here) and obtain a behaviour biased towards a learned visual heading (Fig 10B), mimicking the real behaviour of wood ants in similar experimental paradigm [13]. The unilateral "lesion"/silencing of a MB occurs between the learning phase and the test phase (Fig 9B) and, therefore, blocks the influence of the visual long-term MB memory on the formation of the CX memory (EPG-PFL3 synapses) on the side of the lesion.

When a MB is unilaterally silenced and the agent presented with a landmark at ±45˚ from the feeder, the model reverts to a strong attraction to the landmark in all cases (Fig 10C and 10D). This is due to the importance of both EPG-PFL3 distributions (i.e., one for each side of

the CX model) in defining the orientation in which to keep the EB bump. With only one of the two, the MB memory does not influence the position of the bump and therefore only the innate attraction to the landmark actually influences the behaviour.

## 3 Discussion

Our principal objective was to build a neuroanatomically inspired neural network model [21, 27, 30] capable of explaining the use of the ellipsoid body compass as a reference to maintain the course toward a desired direction [18], reinforced either by innate or learned pathways. We first showed the intrinsic potential of the synapse pattern from the EPG (compass) to PFL3 neurons within the protocerebral bridge, as identified in the *Drosophila* connectome [38], to produce a stable course following the compass bump of activity directly (Fig 5B) or, potentially, with a constant offset (Fig 5C). This is achieved by: (i) the division of the PFL3 neurons into two unilateral subsets independently controlling turns towards each side [38]; and (ii) the unbalanced connectivity scheme between the compass neurons (EPGs) and these unilateral subsets (Fig 4). This organisation corresponds to a more basic, perhaps ancestral, control scheme called tropotaxis [44] in which the left-right comparison of a sensory signal controls the orientation taken [59]. In addition, the control of steering by a continuous probabilistic comparison of left and right turns naturally produces oscillatory or zigzag pattern that resembles paths observed in insects following pheromone trails, odor plumes [60] and/or visual guidance [9, 61, 62]. We show here how this left-right structure of the CX output from the PFL3 neurons could be co-opted by sensory (visual attraction) and memory (MB visual long-term memory) pathways that do not themselves have such a left-right structure, through an adaptively acquired memory. The key mechanism we propose is that an association between self-motion and the compass signal can be acquired under the control of reinforcement. This depends on the known anatomical shift between functional columns in the CX, previously used to support steering output in a path integration model [21]. This shift allows the current compass pattern to be 'mentally rotated' to the left and right, so that the link between the preceding compass pattern, the turning action (right or left) and the consequent reward can be made.

Although in the presented model we have hypothesised that this memory resides in the EPG-PFL synaptic weights, we believe it is equally plausible that the required modification occurs upstream in the FB neurons, which could then have a direct modulatory input to PFL3 neurons (Fig C in S1 Text). Nevertheless, a short-term synaptic plasticity is plausible at the CX level similar to the plasticity observed at the synapses from the Ring neurons to the EPGs [63, 64]; note such short term change in synaptic efficacy may not be observable as changes in the number of synapses [40] (section 2.5). A possibility is that innate behaviours could be imprinted as long-term synaptic differences, modulating the number of synapses [30], while learned behaviour could be supported by more plastic (and reversible change) in the synaptic molecular modulation. Independently from the mechanism of CX memory formation, we demonstrated the suitability of such a memory to support several oriented behaviours; innate attraction to a bar [7, 10], menotaxis behaviour [14] and reaching a memorized location [13].

Such a memory process is coherent with the central role of the CX in spatial learning [65], in the storage of short-term orientation memory [28], and in the convergence of multiple pathways and the selection of a range of sensory driven behaviours [32]. Moreover, this memory process permits considerable behavioural flexibility including permitting orientation even when an offset [15] exists between the compass representation (i.e., the bump location) and allocentric cues in the environment (Fig D in S1 Text). In addition, this model can reproduce the impact of unilateral MB lesions on the behaviour of wood ants [13]; after a simulated

unilateral silencing the model reverts to innate behaviour (under the assumption of a linear addition of the pathways and the unilateral influence of the MB). This emphasizes the importance of both MBs in memory view-based navigation in the CX. It also supports a unilateral influence of each MB on the CX control of the navigation behaviour, also supported by the evidence of lateralization in the insects' memory [66]. As such, there is an interesting correspondence to the model recently proposed by [44] in which the left and right MB are assumed to learn views oriented respectively to the left and right of the target direction. This is comparable to using the unilateral self-motion inhibition signal in our model to select which MB stores the visual memory, rather than to select which set of FBn influence the association of visual memory to reward in the CX. Both models pinpoint the potential role of the CX to integrate and smooth steering driven by the MB visual memory. The model in [44] has the advantage of providing a non-redundant role for the two MBs, whereas our model can be more easily generalised to different modalities, including non-directional reward signals, as shown by the combination of innate and learned behaviour (Fig H in S1 Text).

## 3.1 Assumptions in the model about CX inputs

Our model has abstracted from neuroanatomical reality to simplify the connections from the visual system to the EB, by using a direct retinal input. Indeed, it is known that the visual inputs to the EPGs, conveyed by 'ring' neurons [67], are not fully retinotopic. Ring neurons connect from the Lateral Triangle (LTR) to the EB [34]. LTR glomeruli conserve localized visual receptive fields but form connections with the entire EB tile set [67], potentially modulated by neuronal plasticity [63]. The effect of this connection scheme is the preservation of a visual scenery representation in the EB [63] whereas an arbitrary offset can arise between the external location of visual information and the EB activity 'bump' in bar following experiments [15]. Although we did not explicitly include ring neurons, the model could reproduce the effect of this variable offset through a retinotopic shift in the connection from the visual inputs to the EB, and can function with such offset (Fig D in S1 Text). The representation of innate attraction as a modulatory signal and not as an intrinsic hardwired system allows this flexibility. Alternatively, the apparent randomness of some menotactic behaviour in insects could be explained by a hardwired architecture, where some arbitrary bump offset fixes the preferred direction. The different behaviours (direct attraction or menotaxis) observed in different species and in different contexts [wood ants [7, 9, 13] and flying *Drosophila* [68] attracted by landmark versus walking *Drosophila*, innately [18], and ants, through learning [69], both keeping a landmark in a specific orientation of the visual field] argues for some flexibility in insect oriented behaviour.

The second input we assume is available to modulate the memory is the perception of rotational self-motion. The interaction of optic-flow and compass-mediated signals has already been proposed in the form of a speed measurement for the path integrator [21]. However, in our model, the implementation of a pathway carrying self-rotation estimation is required. The ability of insect visual systems to untangle of the optic flow generated by translation and rotation [70] supports our choice. In addition, self-rotation might also be perceived by a mechanosensory pathway from the legs [71], or be estimated from efference copy from the motor system [72]. Nevertheless, the implementation here is highly inferential; but could be supported by the existence of pathways between the EB and the FB involving the noduli inputs (hAB neurons [42], P-F3N2d and P-F3N2v [39]). In our model, we found that the main advantage of selecting, specifically, ipsilateral inhibition (i.e., right turns inhibit right FBns and left turns inhibit left FBns) was to facilitate the exploration and the integration of several cues (innate and memorized). Contralateral inhibition, for example, would lead to an attraction to

the stronger cue and neglect of sparsely accessible cues such as the MB long-term memory (Fig E in S1 Text).

Finally, a key assumption of the model is that the CX receives a reward signal based on sensory or memory pathways. The FB represents an ideal target to model the influence of parallel sensory pathways on the CX [44] because it receives inputs from several other brain areas, including the MB [73–75], while sharing the columnar organisation observed in the EB and the PB [16]. For example, *Drosophila* FB has been shown to be necessary to the control of the steering in a visual pattern recognition paradigm [76], involving the EB [77]. We did not target a specific neuron type so used a generic FBn denomination. However, the parallel with CPU4a/b identified in bees as part of the path integrator makes the corresponding neurons P-F3N2d and P-F3N2v [39] plausible targets in the FB as they also make mixed synaptic connections to the noduli [39]. In addition, the CX is one of the main structures in the insect's brain, along with the MB, receiving dopaminergic innervation, notably in the FB [78]. The role of dopaminergic input in synaptic modulation and learning processes in the MB [79] suggests a similar role in the CX to generate a synaptic memory. This is consistent with observations that link the dopaminergic circuit in the CX to the regulation to wake/sleep arousal or stimulus specific arousal [80] and more generally its proposed function to modulate the arousal threshold in *Drosophila* [81].

### 3.2 An operant strategy to find where to go

The main principle of our model is the formation of a memory during the exploration of the environment surrounding the insect under the influence of goodness/badness signals derived from innate and/or learned preferences. The exploratory behaviour here is crucial, supporting the scan of the surroundings and integrating the proprioception information correctly (Fig E in S1 Text) to ensure an optimal behaviour. Note that, although our model does not need any systematic scanning of the visual environment to decide the course to take [26], it is not able to provide a steering command to an orientation that has not been previously seen (some other proposed navigation models do have this capacity, e.g., by using the frequency transformation of the visual scenery [22, 82], or a biased memory [44]). However, a large number of insect species express systematic scanning procedures before determining a route/direction to take [e.g, ants [83, 84]; dungbeetles [85], sandhoppers [86]], supporting our model assumptions. To create an exploratory behaviour in our simulation we add noise to the steering output from the PFL3, which is needed to generate correct attraction to the cue or the feeder (Fig F in S1 Text). Adding a stochastic component as noise to the behaviour is intended to account for the effect of locomotor mechanics and/or external disturbances. Alternatively, the oscillatory behaviour observed in the path of insects (wood ants [9], cockroachs [87], locusts [61], wasps [88], moths [89]) could provide a systematic and controlled exploration of their surroundings that would serve the same function. In addition, head movements and initial systematic scanning behaviour [83] offer ways to increase the exploration of the environment while maintaining the course.

### 3.3 Why use the CX for innate attraction?

Our model focuses on the potential of the specific CX architecture to support different oriented behaviours (attraction, menotaxis, visual memory guided) based on inputs in the form of (undirected) attractiveness and/or repulsion. As such, the EB is assumed to play a role even in the innate attraction to a visual landmark, which we assume induces a reward signal when appearing in the frontal area of the visual field, modulating the EPG to PFL3 synaptic weights to cause steering towards it. In walking *Drosophila*, disrupting the EB abolishes menotaxis but

not direct attraction to a cue [18]. Indeed, it seems highly likely that alternative direct sensory pathways for innate attraction exist, such as observed connections from visual lobes to steering neurons downstream from the CX [30], and are able to control the behaviour without requiring the orientation representation provided by the CX [90]. However, although CX memory may not be necessary for attraction to conspicuous cues, it might still provide several advantages in enhancing innate behaviours [43].

One is the creation of a stable heading that can persist even with a disappearance of the cue (s) (Fig 8), as long as the EB bump motion is supported by other cues (for example the proprioceptive cues maintaining heading tracking in darkness, Fig 3D). The existence of a persistent influence of a cue after its disappearance has been shown in *Drosophila* [28, 91], as well as the relative reliability of the EB compass in darkness [15]. Therefore, the CX could act as a smoothing filter, in addition to direct taxis mechanisms, to generate a short-term memory of the sensory world when this appears to be unsteady. For example, the experiments by Neuser et al. [28] on the short-term memory of landmark guidance with disappearing bars support the interplay between sensory taxis pathways and the formation of a short-term orientation memory. In their paradigm, *Drosophila* show a sequence of behaviour while the environment is modified: (i) a patrol behaviour between two bars at 0 and 180˚; (ii) attraction to a new bar appearing at 90˚ while the others disappear; and (iii) a reversion to the patrol behaviour in the original direction, in the absence of any cue, when the newly appeared bar disappears. This would be consistent with our model if it was assumed that the CX memory formed during (i) is over-ridden, but not altered, during the brief presentation in stage (ii), and will guide the behaviour in (iii) when all landmarks disappear (c.f. Fig 8).

Note that these advantages of smoothing and persistence should also apply to enhance either menotaxis or memory-guided orientation over a direct taxis system, e.g., smoothing the integration of the MB signals that can be unsteady [44]. The integration of a sparsely accessible sensory signals could be particularly interesting considering the cluttered and dynamic environments within which insects forage. The ability of desert ants to perform homing while walking backwards would also be supported by a short term storage of the direction indicated by visual memory, which can be set during intermittent 'peeks' to check the direction [92]. In addition, the temporal dynamics properties of neurobiological circuits involved in the smoothing/filtering of this signal probably should include some forgetting mechanisms allowing the memory to be reversible. For the paradigm we used, with a distant landmark for an agent confined to an arena, forgetting was not necessary, but in a more complex navigation scenario where the insect may pass the landmark, forgetting, and therefore the creation of a real working memory, would be crucial. We plan to investigate in the future how this model could be applied to complex route learning/following using an integration/decay dynamics [44], or whether the integration of the path could erase the memory along a route [21].

The temporal integration at the CX level also makes it possible to explore several innately attractive features of the environment and build-up a global direction vector combining them. This individual consideration of landmarks rather than the deciphering of the global layout of the environment would match some observations in insects [93, 94]. It is also noteworthy that attraction to vertical lines in wood ants has been proposed [95] to be sustained by their functional potential to resemble tree trunks (a key source of food) and therefore could be determined by earlier life experience. That is, the nature of the attractive target may become refined as particular visual stimuli are discovered to be rewarding. This can be considered as a more general principle, in which a 'map' of the visual environment's attractiveness [96] generates a heading vector, for any location, based on the EB bump. The same architecture could also be generalised to other sensory inputs, such as olfaction or wind sensing (see Fig G in S1 Text for

gradient ascent capability), to generate such a combined heading vector (or a set of vectors) and enhance the robustness of the behaviour (see section 3.4).

### 3.4 Optimal sensory integration

In addition to its spatio-temporal properties, the convergence of several sensory pathways onto a single compass-driven scheme offers an interesting substrate to look further into optimization of the navigation behaviours. Optimal integration in biological systems has been considered to conform to Bayesian-like principles [23, 24, 97–99], as a principled way to take noise and uncertainty into account. Social insects and central-place foragers have evolved under high pressure to be able to return reliably to their nests, and hence make use of multiple redundant mechanisms, and a wide range of sensory cues including visual cues (polarised light, landmarks, sun, light gradients), magnetic fields, optic flow, odours, wind direction, tactile cues and/or proprioception [100–103]. The problem of how these are combined for robust navigation has been considered in mathematical models [23] but how it might be realised in the brain of the animal is still unknown [22]. Here, the modulation over time of the weight distribution during the exploration of the sensory environment offers a way to generate directly a combined probablility distribution from multiple inputs. An example that we observe from our simulation is the impact of negative rewards in addition to positive ones in creating the landmark attraction (Fig 7). It has been recently shown in homing ants that memorizing anti-views or repellent memories, in addition to an attractive memory, could help in determining the course to take [2, 104]. The combination of these two opposed pathways on our synaptic memory model would automatically create a combined distribution (Fig H in S1 Text), improving the homing performance as proposed in [105].

Importantly, this use of two different reward signals could be directly extended to two (or more) rewards that come from different sensory systems. For example, if the agent was equipped with mechanosensors similar to insect antennae, we could define the equal rearward deflection of both antennnae as rewarding, corresponding to an upwind heading direction. This could be combined with the existing reward when the visual landmark is centred, so that the two inputs simultaneously affect the adaptation of the weights. The relative strength of the two cues might then allow smooth transition between upwind flight when the visual stimulus is weak (far away from a learned location) to visual attraction as this cue becomes stronger or more reliable (from close range). In the model presented here, there is one set of weights between the EPG and PFL3, which can be influenced by multiple reward signals as just described. However, an alternative implementation (with similar computational effect) would be to assume the FB connections provide pre-synaptic modulation of the EPG-PFL3 connections, and the extent of this modulation is under adaptation by the reward signal. This alternative opens up the interesting possibility of having multiple sources of compass information integrated as parallel streams, i.e., several sets of FBns that, instead of copying the EPG compass activity, have an activity bump driven by independent directional cues, such as wind [106–108], or celestial information [101]. Indeed, the summation of several sensory compasses offer a good basis for theoretically optimal cue integration [22]. The combination of different compasses under a single reward provided by sensory and/or memory pathways might allow their flexible re-use depending on their availability and their reliability in the local environment.

Here we combined two different systems for visual reward, one innate and one learned. The interaction between innate and learned modalities is important for insect navigation [7, 102]. We considered the simplest case where the influence of each pathway is added linearly. The exploration of different integration strategies could help us better understand the

interaction between innate and learned behaviours. We therefore believe that this model will help in exploring the ontogeny of the transformation from innate to learned landmark use [7, 109].

## Supporting information

**S1 Text. Supporting methods and figures. Fig A. Simulations with no modulation of the EPG-PFL synapses. A.** 5 examples of paths where the EPG-PFL3 synapse weights are set equal and constant. **B.** Boxplot (Median: Red, Inter-quartile: shaded box) of the exploration ratio, calculated as the percentage of the 360° surroundings faced by the agent, during 50 simulations. **C.** Heatmap of 50 simulations stacked adjusted to the cue location (0°). **D.** Probability density function of the angular speed (all 50 simulations data stacked) corresponding to the 10° s.d. gaussian noise applied to the output of the steering model (red line). **Fig B. Innate attraction under the control of a visual reward signal.** Simulations of the FB steering model (Fig 6) using a reward signal provided by the visual processes to modify the EPG-PFL synapse weights. We created the visual input signal to the CX using different masks. Results for each panel include (a) the final path directions (n = 50 simulations), (b) the probality density function of the final direction of the 50 simulstions, (c) the averaged EPG-PFLs synapse weights and (d) examples of 5 simulation paths. **A.** Visual input to the FBs is equal to the sum of the visual units signal through a continuous proportional mask from 0 (rear units) to 1 (frontal units). **B.** Visual input to the FBs is equal to the sum of the visual units signal through a continuous proportional mask from -0.5 (rear units) to 0.5 (frontal units). **C.** Visual input to the FBs is equal to the sum of the visual units signal through a discrete mask equal to 0 (outside the 30° frontal area) or 1 (inside the 30° frontal area). **D.** Visual input to the FBs is equal to the sum of the visual units signal through a discrete mask equal to 0 (outside the 30° frontal area and the 30° rear area), 0.5 (inside the 30° rear area) or 1 (inside the 30° frontal area). **Fig C. Alternative memory model. A.** Model diagram. The memory is integrated as a separate set of neurons (similar to CPU4 in [21]), which receive inputs from the reward signal (Vin or MB), the self-motion and the EPGs. The inhibition from FBns/CPU4s to the PFL3s/CPU1as present the same shift of 1 column as the model described in Fig 6. **B.** Results from simulation where the reward signal is provided by the visual system and generate an innate attraction to the cue (discrete positive mask). **C.** Results from simulation where the reward signal is provided by the MBs after an initial learning phase. In this case the MB is a singular structure providing a similar signal to each side of the CX model. **Fig D. Using an offset compass does not compromise the model function. A.** The connectivity between the visual units and the E-PGs has been shifted by a random value. The bump produced therefore expresses a constant offset different for every simulation. **B.** Final direction vectors relative to the cue for 20 simulations. **C.** Overall EPG-PFL synapse weights. The lines indicate the mean value for each EPG-PFL couple (red for the right side and green for the left) and the shaded area the standard mean deviation (s.e. m.). **D.** 4 individual experiment paths and the associated EPG-PFL synapse weights generated during the simulation (red for the right side and green for the left). **Fig E. Impact of the self motion integration in the model. A-B-C.** Model simulations with a combination of innate attraction and visual memory (MBs) as presented in Fig 10A & 10B. (a) Final direction vectors for 50 simulations. The arc represent the median (dot) ± 95% C.I. obtained via bootstrap (rep = 10000). (b) Averaged right (red) and left (green) EPG-PFL synapse weights (shaded area: ±*s.d.*) obtain during 50 simulations. **A.** Ipsilateral inhibitory circuit from the self-motion signal to the FBns. This corresponds to the circuit presented in Fig 6. **B.** Contralateral inhibitory circuit from the self-motion signal to the FBns. **C.** Circuit without any integration of the self-motion by the FBns layer. **Fig F. Impact of the steering noise on the model performance.**

The influence of the noise standard deviation applied to the steering is tested in the case of the innate attraction alone ($\omega_{MB} = 0$) and combined with the visual memory ($\omega_{Vin}$ and $\omega_{MB}$ set as in Fig 10B). **Fig G. Gradient ascent properties, potential for the olfactory pathway. A.** The reward signal ($I_{CX}$, input to the CX) is defined here as the difference ($\Delta s$) of the concentration in sensory input [$s(t)$]. Negative input, decrease of the concentration are not considered here ($\Delta s \leq 0 \leftrightarrow \Delta s = 0$). The value of $\Delta$ is multiplied by a free parameter $\omega_{Olf}$ to adapt roughly the amplitude to the same range as the visual pathways (innate and learned). During the simulations the visual compass is maintain, and keep it's function to modulate the EPG-PFL3 synaptic weights, thanks to a landmark randomly positioned around the arena. **B.** Simulations with 3 different sources positioned at 100, 150 and 200 lu. from the arena centre creating a gaussian gradient (respectively s.d. = 50, 100 and 150 lu.). Left panels show the gradient shape and 5 paths example (white lines). Right panels show the final direction vectors of the full set of simulations (n = 20). **Fig H. Influence of an attractive-repulsive MB views memory on the model. A-B-C.** (a) Final direction vectors of 50 simulations. The red arc indicates the median (red dot) 95% C.I. obtained by bootstrap ($n_{rep} = 10000$). (b) 5 path examples. (c) Averaged right (red) and left (green) EPG-PFLs synapse weights (shaded area: $\pm s.d.$). For these simulations, in contrary to the other results presented in the paper, synapses weights were not capped to 0.8. **A.** Simulations with attractive views memory only acquired facing ($\pm 15°$, blue span) the feeder (green dot). This correspond to the same simulations as presented in Fig 9. **B.** Simulations with repulsive views memory only. MBON value is multiplied by -1 before connecting the FB neurons. Repulsive views are acquired in the same manner than the attractive one while facing $180° \pm 15°$ (red span) away from the feeder (green dot). **C.** Simulations with both attractive and repulsive views memory. Each memory use the very same subset of KC neurons but is processed on a different MBON (positive for attraction, negative for repulsion). **D.** Probability density function of the final directions for 1000 simulations in the 3 conditions (Attractive memory, positive memory and attractive & positive memory).
(PDF)

**S1 Table. S1_Table.csv**: Table referencing the list of synaptic connection from EPGs to PFL3s. *Presynaptic_Neuron: ID of the presynaptic neuron; PreN_ID: ID tag of the presynaptic neuron; Postsynaptic_Neuron: ID of the postsynaptic neuron; PpstN_ID: ID tag of the postsynaptic neuron; Nb_Synapses: Synapse count.* **EPG2PFL_connection.txt**: txt version of the reference table. **EPG2PFK_table_connections.csv**: csv file containing the synaptic connection from EPGs to PFL3s as a cross table. **TableSynapses.pdf**: image plot representing the extracted cross table synaptic connection from EPGs to PFL3s. **WeightCircular.pdf**: circular plot conversion to the 1-on-1 EPG to PFL3 synaptic weights.
(ZIP)

## Acknowledgments

We are very thankful to Ioannis Pisokas for the discussions on the EB modeling and to James Knight for his comments on the manuscript.

## Author Contributions

**Conceptualization:** Roman Goulard, Barbara Webb.

**Data curation:** Roman Goulard.

**Formal analysis:** Roman Goulard.

**Investigation:** Roman Goulard.

**Methodology:** Roman Goulard.

**Supervision:** Barbara Webb.

**Validation:** Barbara Webb.

**Writing – original draft:** Roman Goulard.

**Writing – review & editing:** Roman Goulard, Cornelia Buehlmann, Jeremy E. Niven, Paul Graham, Barbara Webb.

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
