## [Decision Letter · Decision Letter 0]

5 May 2021

Dear Dr GOULARD,

Thank you very much for submitting your manuscript "A unified mechanism for innate and learned visual landmark guidance in the insect central complex" for consideration at PLOS Computational Biology.

As with all papers reviewed by the journal, your manuscript was reviewed by members of the editorial board and by several independent reviewers. In light of the reviews (below this email), we would like to invite the resubmission of a significantly-revised version that takes into account the reviewers' comments.

We cannot make any decision about publication until we have seen the revised manuscript and your response to the reviewers' comments. Your revised manuscript is also likely to be sent to reviewers for further evaluation.

Sincerely,

Joseph Ayers, PhD

Associate Editor

PLOS Computational Biology

Lyle Graham

Deputy Editor

PLOS Computational Biology

Reviewer's Responses to Questions

**Comments to the Authors:**

Reviewer #1: Goulard and colleagues developed an insect central complex model to explain how the central complex might combine innate and learned orienting behaviors. It describes how non-navigational information, such as valence, could be used to modify synaptic weights to affect navigation behaviors.

The main contribution of this work is to provide a conceptual framework of how reward signal can be used to modify the synapses from head direction cells to motor command cells. The critical part is the introduction of the shifted modulation (figure 6), in columns, to generate stable orienting behaviors. This conceptual contribution is potentially important. However, there are a few concerns that need to be addressed.

1. [page 9] This work singularly picked PFL3 as its key player of the model, in which PFL3 neurons directly receive the input from EPG neurons via synapses in the PB. On the other hand, the hypothetical set of neurons providing modulatory signal are modeled to be in the FB and they modify the synapses in the PB in shifted columns, an essential assumption of the model. I am not sure if this is anatomically (or even biologically) feasible or supported. Although the conceptual contribution of the model is still very interesting, based on anatomy, it is hard to imagine that PFL3 neurons (and the hypothesized FB neurons) are responsible for the proposed mechanism.

2. There is no evidence that the synapses from EPG to PFL3 are plastic.

3. Related to above: The synapse data from EM in Figure 4 shows strong symmetrical bias in left and right connections. It is not clear if this kind of strong anatomical structure can be achieved in a short-term implied in this work. If synapses are plastic in a short-term, the actual synaptic weight might be modified not in the number of synapses but rather in some other mechanisms (such as the postsynaptic receptor concentration/distribution). For example, the synapses between Ring neurons and EPG neurons have been hypothesized to be plastic (Kim et al., 2019; Fisher et al., 2019), yet the EM data shows nearly homogeneous number of synapses between them (Hulse et al., 2020).

4. From eq (4), the synaptic weights may not be stable if the simulation is continued over multiple scenarios. For example, if the reward mask in the front moves slowly counterclockwise over time, while the plasticity is ON, then the entire weight would be eventually saturated during the simulation. In other words, the synaptic update is only one-way, which, without a ‘forgetting’ mechanism, may result in unstable agent behaviors.

5. line 131-132: It says the connection from EPG to PFL3 is inhibitory since EPGs are cholinergic, which means excitatory. As the polarity of the connection from EPG to PFL3 is critical for the model behavior, this discrepancy needs to be clarified.

6. The description of figure 6 and the eq (4) is arguably the most important part of the paper, yet it was not easy to grasp the key idea. I strongly suggest more polishing.

Minor comments

- A lot of subpanels of figures are not referenced in the main text, which may leave readers confused. Figure 1 is not even mentioned in the main text.

- Line 17: Ref [17] does not seem a right one to cite. Taube et al 1990 that found HDCs seems more appropriate.

- The edge index used to calculate the output of visual neurons is hard to understand its behavior and meaning. The text says that ‘S_left’ and other three ‘S’s are the summed intensities of the corresponding half. Then what does I_i =0 (and I_i=1) mean? In my calculation, I_i would be 1, only if the bright pixels are only on the left top corner, no matter how the edge shapes. On the other hand, if there is a vertical edge at the center of the scene, I_i=0.75. Please clarify what I_i really means.

- Table 1 in the supplementary info needs to be sorted.

- Table 1 in the supp info and Figure 4 shows 7 PFLs on each side, whereas the text says there are 12 PFLs for each side. Is there a reason for this discrepancy?

- line 266-276: It is not clear whether the modification of the synapses from EPG to PFL3 happens after the lesion, or the lesion was performed after the learning. I assume the former, since the reward signal, which is used to modify the synaptic weight, is affected by the lesion; It needs to be clarified.

- line 378-386: The relationship between Neuser at al. and this work is not clearly described in the discussion. (Is it future work? Or can this model replicate it? Or what kind of mechanisms are necessary to replicate it?)

Reviewer #2: The authors build a neuroanatomical model including head direction neurons in the central complex to produce observed innate attraction to visual clues. They integrate the mushroom bodies into the model, which they used to encode visual memory and to enable memory-based directional navigation in a simulated agent. The core idea of the model is interesting and relies on the connectivity from EPG to PFL3s neurons, which the authors found to be heterogeneous from the fly connectome (Figure 4b). Since PFL3s neurons output to previously reported DaN2 neurons involved in steering control [Wilson lab, biorxiv (2020)], the authors argue that these heterogeneous connections from EPG to PFL3 produce fixation towards a desired direction.

One issue that is still puzzling is the assumption that EPGs are inhibitory to PFL3, which seems not to be supported by experimental evidence. This also seems to be at odds with another assumption: that EPG neurons are excitatory to PEN and PEG neurons.

The polarity of EPG neurons in the model should therefore be clarified. It seems that assuming EPG to be excitatory to PFL3 neurons would result in the agent going away from the visual clue.

A second issue that would be good to improve is some biological justification for the plasticity, since these rules are not standard.

Comments:

- Line 132: “EPG are reported to form cholinergic synapses”: citation missing.

- Line 132: The authors assumed in equation (2) that EPG provide excitatory input to PEG and PEN neurons, while now they assume EPG neurons provide inhibitory input to PFL3 neurons. EPG are reported as excitatory for example in [Franconville, R. …, Jarayman, V. Elife (2018). Elife], where functional connections involving presynaptic EPG neurons are always excitatory.

- Line 132: Missing equation for activity of PFL3 neurons

- Line 145: If the model assumes excitatory connections between EPG and PFL3, then the agent would not be attracted to the stimulus but would avoid it and position the stimulus in the back.

- Figure 5B: The direction of the agent in the absence of Gaussian noise is fully determined by the connectivity and the preferred direction will be defined by the intersection point between the left and right connectivity profiles of EPG-PFL (intersection between green and red line in Figure 5B, top left). It would be helpful, if the authors explained this in more detail, and then added the noise that produces stochastic trajectories.

- Line 165: Some neurons in the FB perform column-shift operations (for example, see hAB neurons in [Cheng L. …, Maimon, G. Biorxiv (2020)] ). Why not include a population from the fly connectome that matches the population ‘FB’ that the authors introduce?

- Equation (4) is an unusual plasticity rule. It has 2 modulatory signals that have to coincide in time: activity of FB(t) neurons and a reward signal Rew(t). It would be helpful to add more biological justification for this rule.

- Line 204: When the model uses negative input, the performance increases. Missing a qualitative measurement for performance to compare between different inputs.

- Line 205 (Figure instead of ‘igure’)

- Figure 7 top row: What is the irregular black semitransparent shadow? Is it the mask? If so, why does it have such an irregular shape?

- Equation (7) does not include time dependence as the previous equations.

- Table 1 seems unnecessary. Why not write in equation (7) the weights w_i^KC-MBON in terms of the activities of DAN and KC:

w_i^KC-MBON = w_i^KC-MBON – 0.2 * DAN * KC

Still this synaptic rule needs biological justification.

- Line 247: During the 100 first steps of the simulation, the agent scans the environment from -15 to 15 degrees. How is this exactly done? During these 100 first steps, are the DAN neurons set manually to 1 so they create the memory? In that case this is a training phase. If after the 100 first steps, the agent is let alone in the environment with activity of DAN neurons set to 0, that is a test phase. Need to clarify (and maybe add a figure) about the timeline and different phases during the simulation.

- Line 252: Needs more justification and a mathematical expression of how the input signal from MB to the CX is ignored in the model. Add to Methods or supplementary information.

- Line 265: What is the evidence that the two MB structures (located in each hemisphere) get visual input from both eyes?

- Discussion could also be shorter, some of the text could go to the introduction, for example most of section 4.3

Reviewer #3: This is an interesting and timely theoretical study in the field of spatial navigation. The authors have used the most up-to-date experimental and anatomical data to construct a nice model of the compass-guided steering system in the insect brain. The basic steering model (Figure 5) is similar to that of Rayshubskiy et al. (ref. 30). The novel aspect of this manuscript is primarily the further extension of that basic model, as shown in Figures 6-8. In this extension, the authors show how the system can learn to adopt a new heading goal associated with some rewarding cue. This learning process relies on a specific hypothetical class of fan-shaped body neurons (called here FBn) which combine heading signals and steering signals. The output of these FBn neurons is used to drive associative plasticity at compass neuron output synapses. This in turn changes the way that the compass is “read out” by the steering system.

Major points:

1. 131 – “The synapses between EPGs and PFL3s are represented as inhibitory (Figure 5A) as EPGs are reported to form cholinergic synapses.” – Cholinergic synapses are excitatory, not inhibitory. Thus the synapses between EPGs and PFL3s should be excitatory. I think this will flip the stable fixed point of the model to the opposite side of the ellipsoid body. I think it will also necessitate an inversion of the learning rule in Equation 4.

2. In addition to the sign inversion mentioned above, I think there must be another sign inversion somewhere in the model. This is because the stable fixed point of this system centers the EPG bump on glomeruli 5R/5L (as shown by Rayshubskiy et al.), but when EPG-to-PFL3 synapses are set to be excitatory, this should flip the stable fixed point of the model. Thus, I think some other sign inversion must be made to restore the correct fixed point. I can’t figure out exactly what this sign inversion is, but the authors may be able to determine this. It may have to with how heading direction changes are mapped onto the EPG array.

3. A key feature of the model is that EPG synapses onto PFL3_left and PFL3_right neurons can be differentially strengthened during right- versus leftward turns. The source of this direction-selective turning-related input is some abstract cell type in the FB (“FBn”). I understand that this is a conceptual model, and the specific identity of the cell in question does not really matter for the larger conceptual point. But if the model has any relevance to the brain, then there should be at least some conceivable pathway where this direction-selective turning-related input could originate. The authors suggest this input comes from the noduli, which seems like a good guess. If so, they should be able to show that a given PFL3 neurons receives mainly input from either the left-nodulus OR right-nodulus. Of course, these connections may be indirect, but still it should be possible to show that there is differential indirect anatomical input from the left-nodulus or right-nodulus onto any given PFL3 neuron. I am not asking the authors to do any experiments – I am just asking them to look at the connectome and point out a pathway that could make this idea plausible.

4. I am confused as to whether the FBn inputs to PFL3 neurons (Figure 6) are active all the time (during all navigation), or only during learning. If they are active all the time, wouldn’t they tend to truncate and reverse any turning behavior – because rightward turns cause an immediate withdrawal of excitatory drive to the PFL3 neurons that drive rightward turning, and an increase in excitatory drive to the PFL3 neurons that drive leftward turning? If FBn inputs are postulated to be inactive except during learning, then how is this thought to work, and how is learning separated from navigation per se?

Minor points:

5. 94 – “mimicking bar fixation experiments in Drosophila” – Bar fixation does not require the central complex (see Green et al. 2019 ref. 18 and Giraldo et al. 2018 ref 19). Therefore this sentence is potentially confusing and should be omitted. As noted in the next sentence, fixation is not the task which is being modeled here anyhow; rather, the task is to hold the visual cue at an arbitrary angle.

6. Figure 3B, regarding the retinotopic mapping from visual neurons to EPG neurons: this is actually a likely accurate description of the functional mapping from Ring neurons to EPG neurons. See for example Kim et al. Figure 2e (ref 57) and Fisher et al. 2019 Nature Figure 5g (https://doi.org/10.1038/s41586-019-1772-4).

7. Figure 3C: It should be noted explicitly in the legend that the compass is here depicted as viewed from the posterior side of the brain. If the compass is viewed from the posterior side, then the bump moves counterclockwise as the fly turns right, and vice versa ((Turner-Evans et al., 2017).

8. 136 – “More specifcally, the specific weights/connections pattern (Figure 5B) that frames the EB bump to the front of the visual field would sufice to generate an innate attraction to the conspicuous landmark.” – See major point 1 above. This should be reversed: front->back (or attraction->repulsion).

9. 159 – “in our case the motion input corresponds to left and right rotations rather than translation” – It seems worth noting here that sideways translation is highly correlated with rotation, and so it would be mathematically impossible to be highly correlated with one but not the other.

10. 168 – “primarily motivated by the permanent modification that could define an innate preference for vertical bars”. The innate preference to fixate (center, approach, etc.) a vertical bar does not require the central complex (see Green et al. 2019 ref. 18 and Giraldo et al. 2018 ref 19).

11. Re: section 3.7, it seems relevant to note that there are several MBONs that make an unusually large number of synapses onto FB tangential cells (Li et al. eLife 2020, https://elifesciences.org/articles/62576).

12. It would be useful to provide Table 1 in machine-readable format.

**Have the authors made all data and (if applicable) computational code underlying the findings in their manuscript fully available?**

Reviewer #1: **No: **I couldn't find any information about the simulation code.

Reviewer #2: **No: **

Reviewer #3: **No: **

PLOS authors have the option to publish the peer review history of their article (what does this mean?). If published, this will include your full peer review and any attached files.

Reviewer #1: No

Reviewer #2: No

Reviewer #3: No
---

## [Decision Letter · Decision Letter 1]

26 Aug 2021

Dear Dr GOULARD,

We are pleased to inform you that your manuscript 'A unified mechanism for innate and learned visual landmark guidance in the insect central complex' has been provisionally accepted for publication in PLOS Computational Biology.

Best regards,

Joseph Ayers, PhD

Associate Editor

PLOS Computational Biology

Lyle Graham

Deputy Editor

PLOS Computational Biology

Please address reviewer 3's issue with regard to the causality of the PFL3 neurons

Reviewer's Responses to Questions

**Comments to the Authors:**

Reviewer #1: All scientific concerns have been addressed.

- The figure 1 is still not referred in the main text.

- I strongly recommend adding detailed comments to the github simulation code. It is not easy to understand the structure of the code. It is also desired to include in "Readme" file the general description of the code organization and directives that helps understanding the code. A pdf with a diagram would be very useful.

Reviewer #2: The authors have answered all our concerns. We didn't quite follow the argument in the two following sections, that would still benefit from some more detailed explanations:

The authors stated in their response:

1)

Line 132: “EPG are reported to form cholinergic synapses”: citation missing.

Citation added.

Could you explicitly say how that information was extracted from the cited references? That seems not entirely obvious, but maybe I just missed it.

2)

'We have added in the discussion two sentences introducing the P-F3N2d and P-F3N2v populations (Franconville et al., 2018) as hypothetical targeted populations as they correspond to the CPU4a/b populations that underly the path integration in Stone et al., 2017. These two populations present the advantage of forming connection from the FBs to the PB as well as with the noduli, which could support the integration of the self-motion in the model.'

Could you explain still in more detail how these populations would do that? Currently this does not become clear from the discussion.

Reviewer #3: 1. Regarding point 1 of my original review: Goulard et al. have postulated in that PFL3 neurons in the right PB promote right turns, whereas PFL3 neurons in the left PB promote left turns (Figure 5). PFL3 neurons in the right PB actually synapse directly onto DNa02 neurons on the left, and vice versa (Hulse et al. 2020). When left DNa02 activity is higher than right DNa02 activity, this produces leftward turning (Rayshubskiy et al. 2020) So, in effect, the authors are assuming that PFL3 neurons are inhibitory. They should say this explicitly. They should also note explicitly that Hulse et al. 2020 also assume that PFL3 neurons are inhibitory, whereas Rayshubskiy et al. assume that PFL3 neurons are excitatory. Readers are liable to become confused unless this is made explicit.

2. The authors cite Lyu et al. (ref 41) as evidence of a left-right shift in PB projections to hDeltaB neurons in the FB, as well as evidence of nodulus inputs to hDeltaB (via PFN neurons). In all these citations, they should also cite Hulse et al. 2020 (https://www.biorxiv.org/content/10.1101/2020.12.08.413955v2) and also Lu et al. 2020 (https://www.biorxiv.org/content/10.1101/2020.12.22.424001v1), as both of these papers have relevance equal to that of Lyu et al.

**Have the authors made all data and (if applicable) computational code underlying the findings in their manuscript fully available?**

Reviewer #1: Yes

Reviewer #2: Yes

Reviewer #3: Yes

PLOS authors have the option to publish the peer review history of their article (what does this mean?). If published, this will include your full peer review and any attached files.

Reviewer #1: No

Reviewer #2: No

Reviewer #3: No

---

## [Editor Report · Acceptance letter]

14 Sep 2021

PCOMPBIOL-D-21-00477R1 

A unified mechanism for innate and learned visual landmark guidance in the insect central complex

Dear Dr GOULARD,

I am pleased to inform you that your manuscript has been formally accepted for publication in PLOS Computational Biology. Your manuscript is now with our production department and you will be notified of the publication date in due course.

With kind regards,

Zsofi Zombor
